# Maternal antibodies facilitate Amyloid-β clearance by activating Fc-receptor-Syk-mediated phagocytosis

Tomer Illouz [1,2], Raneen Nicola[1,2], Linoy Ben-Shushan [1,2,3], Ravit Madar[1,2,3], Arya Biragyn[4] & Eitan Okun[1,2,3 ✉]

Maternal antibodies (MAbs) protect against infections in immunologically-immature neonates. Maternally transferred immunity may also be harnessed to target diseases associated with endogenous protein misfolding and aggregation, such as Alzheimer's disease (AD) and AD-pathology in Down syndrome (DS). While familial early-onset AD (fEOAD) is associated with autosomal dominant mutations in the *APP*, *PSEN1,2* genes, promoting cerebral Amyloid-β (Aβ) deposition, DS features a life-long overexpression of the *APP* and *DYRK1A* genes, leading to a cognitive decline mediated by Aβ overproduction and tau hyperphosphorylation. Although no prenatal screening for fEOAD-related mutations is in clinical practice, DS can be diagnosed in utero. We hypothesized that anti-Aβ MAbs might promote the removal of early Aβ accumulation in the central nervous system of human *APP*-expressing mice. To this end, a DNA-vaccine expressing $A\beta_{1-11}$ was delivered to wild-type female mice, followed by mating with 5xFAD males, which exhibit early Aβ plaque formation. MAbs reduce the offspring's cortical Aβ levels 4 months after antibodies were undetectable, along with alleviating short-term memory deficits. MAbs elicit a long-term shift in microglial phenotype in a mechanism involving activation of the FcγR1/Syk/Cofilin pathway. These data suggest that maternal immunization can alleviate cognitive decline mediated by early Aβ deposition, as occurs in EOAD and DS.

[1] The Leslie and Susan Gonda Multidisciplinary Brain Research Center, Bar-Ilan University, Ramat Gan, Israel. [2] The Paul Feder Laboratory on Alzheimer's disease research, Bar-Ilan University, Ramat Gan, Israel. [3] The Mina and Everard Goodman faculty of Life sciences, Bar-Ilan University, Ramat Gan, Israel. [4] Immunoregulation Section, Laboratory of Immunology and Molecular Biology, National Institute on Aging, Baltimore, MD, USA. ✉email: eitan.okun@biu.ac.il

Maternal antibodies (MAbs), transferred from the mother to the offspring, provide critical and effective protection against pathogens in the immunologically immature neonates[1,2]. Maternal IgG transplacental transfer to fetuses[3] and throughout breastfeeding in neonates[2] is mediated by Fc-receptor-neonatal (FcRn) IgG transport across the placental syncytiotrophoblasts[4] and through the intestinal epithelium[5,6], respectively. Therefore, MAbs may be used to boost neonates' immunity by transferring antibodies (Abs) from vaccinated mothers to the offspring. Maternal vaccination can be administered during pregnancy against influenza infection, Tetanus toxoid, reduced diphtheria toxoid, and acellular pertussis. Pneumococcal, Meningococcal, and hepatitis A and B vaccines may be given in pregnancy in specific populations. All the vaccinations mentioned above can also be administered post-partum, during breastfeeding, or both[7,8]. Thus, in many countries, maternal vaccination is a routine procedure during pregnancy, post-partum, and breastfeeding.

We, therefore, hypothesized that maternal immunity might also be harnessed to target diseases associated with early endogenous protein misfolding and aggregation, such as familial early onset Alzheimer's disease (fEOAD) and Alzheimer's disease (AD)-related pathology in Down syndrome (DS). In contrast to the more common late-onset AD (LOAD), fEOAD is associated with autosomal dominant mutations in the Amyloid-precursor protein (APP) and Presenilin 1 and 2 (PSEN1, 2) genes[9,10]. EOAD commences before the age of 65 and is the most common cause of early onset neurodegenerative dementia[10]. EOAD and LOAD share common features, such as neuritic plaques and neurofibrillary tangles, but differ in onset, severity, and localization of the pathology in the brain[9,10]. In an extreme and rare example, individuals carrying the very aggressive mutation L166P in PSEN1 that induces an exceptionally high increase of Amyloid-β (Aβ)-42 production exhibit onset of AD in adolescence[11].

DS, caused by trisomy of chromosome 21, is the most prevalent genetic cause for intellectual disability, affecting ~1 in every 750 live births[12]. The life expectancy of individuals with DS has risen significantly in the past several decades due to improved medical treatment of various comorbidities. Nevertheless, individuals with DS have a shortened life expectancy and accelerated aging compared to the general population[13]. DS shares common pathological symptoms with AD[14], as individuals with DS experience AD-related cognitive decline in their fourth and fifth decades of their lives[14,15]. A significant and widespread AD-related pathology precedes the cognitive decline, detected as early as at the age of 12 years[16], which includes neurofibrillary tangles, Aβ angiopathy, extracellular Aβ and neuritic plaques[17] and microgliosis[18–20]. This is in part due to overexpression of the APP and DYRK1A genes, located on the triplicated chromosome 21[21], although Aβ deposition is also promoted by non-APP mediated mechanisms[15,22]. In this respect, Aβ-related pathology in DS shows high similarity to EOAD[23]. Thus, Aβ pathology is etiologically and mechanistically shared between EOAD and DS[24–26]. While prenatal screening for fEOAD-related mutation is currently not in clinical practice, DS is typically diagnosed towards the end of the first trimester of the pregnancy. Therefore, individuals with DS, as do individuals with some adolescence-onset variants of fEOAD, such as the L166P in PSEN1, may present unique groups that can benefit from an early intervention directed at Aβ pathology.

AβCoreS is a DNA vaccine coding Aβ[1-11] (a B-cell epitope) fused to a Hepatitis-B surface antigen (HBsAg) and a Hepatitis-B core antigen (HBcAg), both contain multiple T-helper epitopes that help facilitate Ab production[27]. The AβCoreS construct was previously shown to delay cognitive decline and reduce human Aβ pathology in the 3xTg-AD mouse model of AD[27]. A modified vaccine targeting murine Aβ, which is triplicated in the Ts65Dn mouse model of DS[28], was also shown to induce anti-Aβ Ab production that facilitates clearance of soluble oligomers and small extracellular inclusions of Aβ from the hippocampus and cortex of Ts65Dn mice[29]. Anti-Aβ Ab production was correlated with reduced neurodegeneration and restoration of the homeostatic phenotype of microglia and astrocytes in this strain[29]. These findings support the notion that immunotherapy against Aβ can slow the progression of dementia in DS and that an anti-murine Aβ vaccination is safe to use in mice, as vaccinated wild-type (WT) mice exhibited no cognitive or other behavioral abnormalities that could indicate neurotoxicity[29].

In the current study, we hypothesized that maternal vaccination at the developmental and postnatal stages coupled with an active vaccination post-partum would yield a continuous immune coverage against Aβ, effectively targeting early neuropathology and dementia as occurs in fEOAD and DS.

Mouse models of trisomy-21 successfully recapitulate several aspects of DS, such as cholinergic neurodegeneration, elevated levels of Aβ and phospho-tau, and impaired cognitive ability[30]. However, these current DS models critically fail to recapitulate extracellular Aβ plaque accumulation[22,29]. Indeed, several DS models, including the Ts65Dn strain, overexpress murine Aβ, which does not aggregate as human Aβ (hAβ), potentially due to three amino-acids differences in the N-terminus. These differences have been reported to modulate the binding of metal ions, which can influence the fibrillogenesis of Aβ peptides[31]. The replacement of His-13 for Arg in rodent Aβ disrupts a metal coordination site, rendering the rodent peptide less prone to zinc-induced aggregation in vitro. Moreover, this region is essential for the specificity of amyloid interactions[32]. Thus, mouse models that encompass triplication of murine APP exhibit elevated levels of APP protein and soluble Aβ but not plaque pathology. Humanized models of DS such as the Tc1 strain also lack plaque pathology due to a nonfunctional human APP gene[33]. Accordingly, these models lack plaque-associated microglial pathology[29,34,35].

These considerations led us to use the 5xFAD strain that encompasses five mutations associated with Aβ pathology in fEOAD[36]: the KM670/671NL (Swedish), I716V (Florida), and V717I (London) mutations in APP and the M146L, L286V mutations in PSEN1.

Herein, WT female mice were vaccinated against hAβ[1-11] or a sham vaccine at 8w of age. Vaccinated females were then crossed with 5xFAD males to produce maternally vaccinated transgenic and WT offspring. Following weaning, transgenic offspring were actively vaccinated against Aβ[1-11] or sham. The offspring were then tested for cognitive capacity at 4 m and were sacrificed for neuropathology assessment at 5 m. Aβ-specific antibodies from vaccinated mice directly affected the phagocytosis capacity of primary microglial cells in a Syk-dependent manner. Our findings provide evidence for a long-term effect of maternal vaccination on Aβ-related pathology, long after maternal Abs were not detectable in the offspring's circulation.

## Results

**Maternal anti-Aβ antibodies are transferred to 5xFAD fetuses and newborns.** Female C57BL/6j mice (8w-old) were immunized against Aβ[1-11] using the AβCoreS-DNA construct or HBsAg as control (Fig. 1a). Vaccinated females produced a high anti-Aβ Ab titer compared with controls ($P < 0.0001$, Fig. 1b). Both mouse groups generated anti-HBsAg titers. However, mice vaccinated with the HBsAg alone produced a higher titer than mice vaccinated with AβCoreS that includes Aβ[1-11] and HBsAg domains ($P < 0.0001$, Fig. 1c), possibly due to epitope competition. This is further indicated by a negative correlation of $r = -0.5$ found between anti-Aβ and -HBsAg Ab levels ($P < 0.05$, Fig. 1d). The AβCoreS vaccine elicited high IgG1, IgG2a, IgG2b, and IgM titers

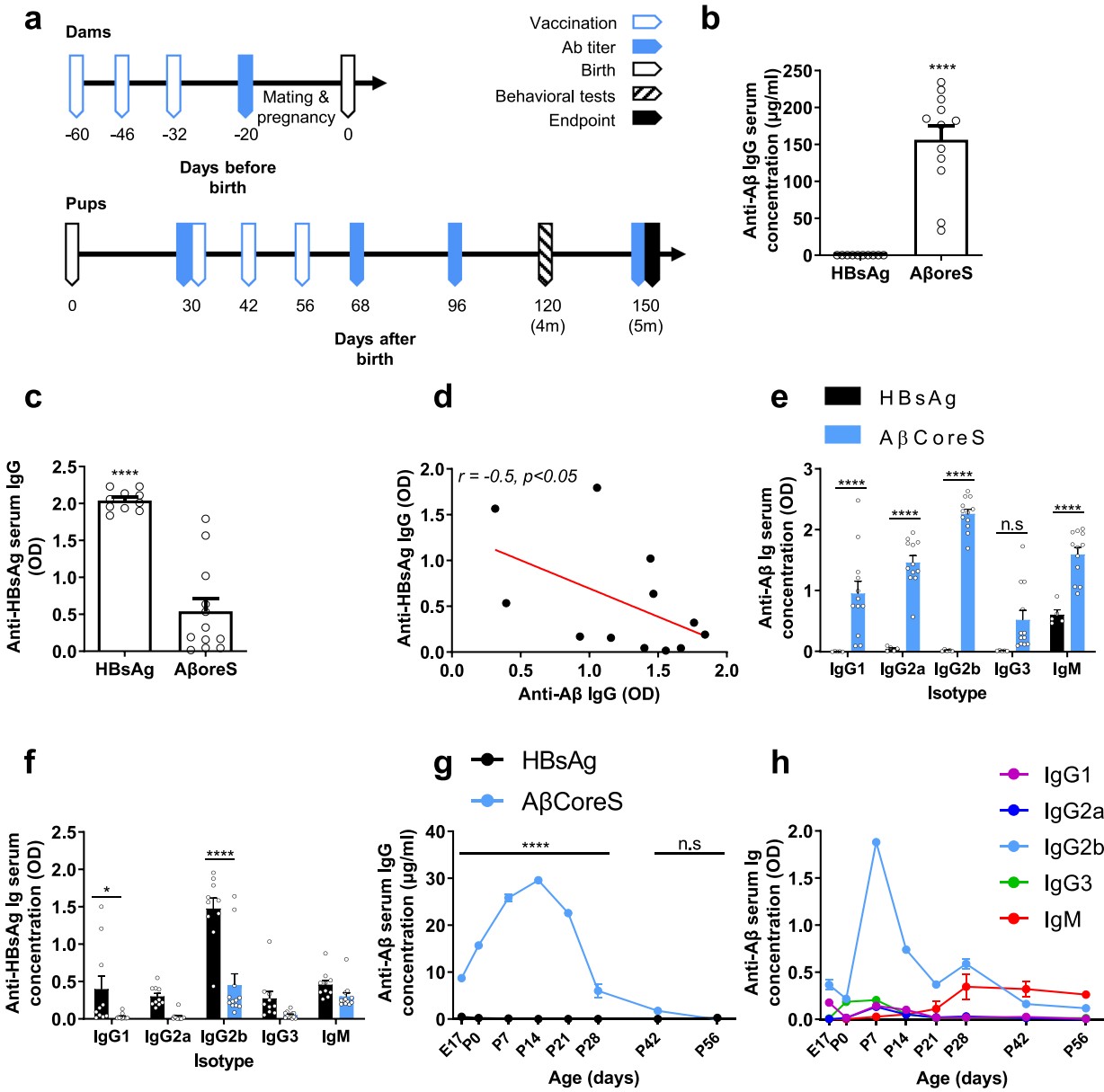

**Fig. 1 Maternally induced anti-Aβ antibodies cross the placenta and lactation into the circulation of 5xFAD fetuses and newborns. a** Study design for dams (upper timeline) and pups (bottom timeline). Serum anti-Aβ antibodies were measured in vaccinated dams (**b–f**), fetuses, and newborns (**g–h**). **b** Anti-Aβ IgG titer in AβCoreS (*n* = 12) and HBsAg (*n* = 10) vaccinated dams. **c** Anti-HBsAg IgG titer in AβCoreS and HBsAg vaccinated dams. **d** A negative correlation between anti-Aβ IgG and anti-HBsAg IgG in vaccinated dams. **e** Anti-Aβ Ig isotypes in AβCoreS and HBsAg vaccinated dams. **f** Anti-HBsAg Ig isotypes in AβCoreS and HBsAg vaccinated dams. **g** Maternally induced anti-Aβ Abs cross the placenta and lactation to the circulation of fetuses and newborns. **h** IgG2b is the most transferable isotype via the placenta and lactation. *$P < 0.05$, ****$P < 0.0001$, unpaired *t*-test, linear regression, Pearson's correlation, one-way ANOVA, two-way ANOVA, repeated measures two-way ANOVA, data are presented as mean ± SEM.

against Aβ ($P < 0.0001$, Fig. 1e). HBsAg alone induced higher IgG1 levels ($P < 0.05$, Fig. 1f) and higher IgG2b levels ($P < 0.0001$, Fig. 1f) compared with AβCoreS-vaccinated mice.

Following vaccination, WT females were mated with 5xFAD males to produce hemizygous transgenic and WT offspring. For Abs analysis, circulating blood was taken from 5xFAD fetuses at E17, P0, P7, P14, P21, P28, P42, and P56. Maternal Anti-Aβ Abs crossed the placenta to the fetuses, as early as E17, compared with controls ($P < 0.0001$ Fig. 1g). Strikingly, these Abs crossed to a greater extent via lactation of the colostrum (P0) and milk (P7-P21). Ab levels gradually increased in vaccinated fetuses compared with controls, with the progression of breastfeeding reaching a peak at P14 ($P < 0.0001$, Fig. 1g). Following weaning at P28, Ab

levels decreased to those of controls. At P42, no Ab difference was observed between the two groups ($P = 0.12$, Fig. 1g). IgG2b was the most prevalent immunoglobulin (Ig) isotype to transfer via the placenta and lactation, peaking at P7 (Fig. 1h).

**Active immunization of maternally vaccinated 5xFAD pups induces mostly anti-Aβ IgG2b Abs.** We next assessed whether maternal vaccination is comparable to active vaccination at a young age. Prior to active vaccination, elevated levels of Aβ-specific IgG2b and IgM were found in maternally vaccinated pups compared with controls ($P < 0.0001$, Fig. 2a). Normalizing the pups' isotype concentration to those of their dams revealed that

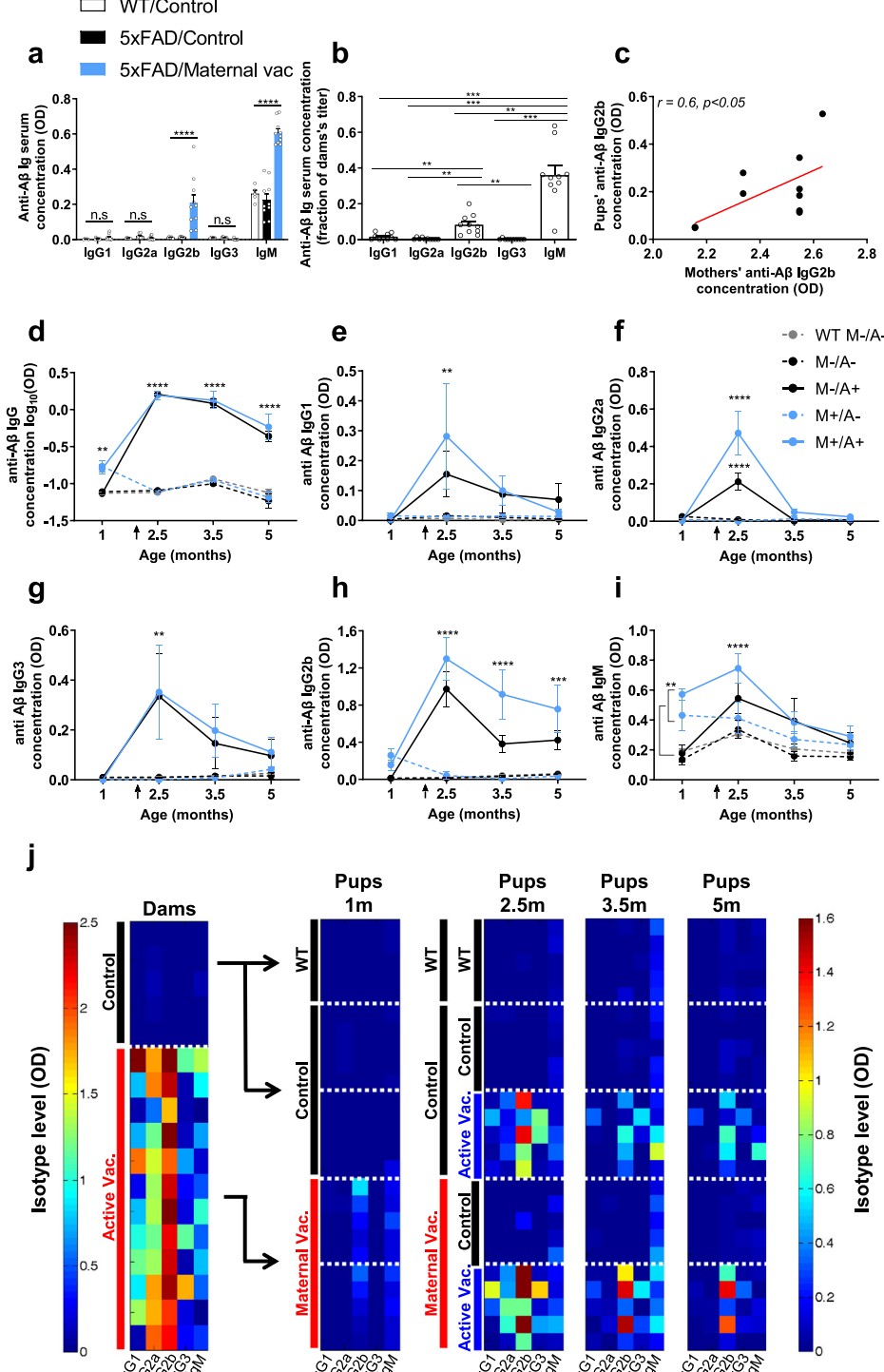

**Fig. 2 Active immunization induced high anti-Aβ antibody production in maternally vaccinated 5xFAD mice.** Serum anti-Aβ antibodies were measured in the offspring following maternal vaccination at 1 m of age (**a**–**c**) as well as following active vaccination (**d**–**j**). **a** Antibody titer was measured in the serum of maternally vaccinated and control 1m-old mice. **b** Ig isotypes in the serum of 1m-old offspring, normalized to mothers' titer before pregnancy. **c** Offspring IgG2b is positively correlated with maternal IgG2b prior to pregnancy. Maternally and sham-vaccinated mice were actively vaccinated at 1 m of age against Aβ (*n* = 5). Antibody titer was measured at 2.5, 3.5, and 5 m. **d** log₁₀OD of total IgG concentration in the serum. Serum levels of (**e**) IgG1, (**f**) IgG3, (**g**) IgG2a, (**h**) IgG2b, and (**i**) IgM. **j** Color-coded heat maps of IgG isotypes in vaccinated dams prior to mating (left panel), isotypes in maternally vaccinated offspring at 1 m of age, and following active vaccination at 2.5, 3.5, and 5 m of age. **\*\****P* < 0.01, **\*\*\****P* < 0.001, **\*\*\*\****P* < 0.0001, linear regression, Pearson's correlation, one-way ANOVA, two-way ANOVA, repeated measures two-way ANOVA, data are presented as mean ± SEM.

IgG2b levels were 8.3% of the mothers' circulating concentration, significantly higher than IgG1, IgG2a, and IgG3 ($P < 0.01$, Fig. 2b). A significant positive correlation of $r = 0.6$ was found between dams' and pups' IgG2b levels ($P < 0.05$, Fig. 2c). Normalized IgM levels were also high in pup circulation, reaching 36% of the mothers' concentration, higher than all other isotypes ($P < 0.01$, Fig. 2b).

At 1 m of age, both groups of $A\beta_{1-11}$ and sham maternally vaccinated pups received either $A\beta_{1-11}$ or sham active vaccination, to create four experimental groups: No treatment (M−/A−), maternal vaccination only (M+/A−), active vaccination only (M−/A+) and combined maternal and active vaccination (M+/A+). A control WT group was administered with the control vaccine only for behavioral and neuropathological baseline (Fig. 1a). Blood Ab titers at 1 m reflected maternal IgG, whereas Ab titers at 2.5, 3.5, and 5 m reflected a humoral response to active vaccination. Following active vaccination, total IgG levels in the M−/A+ and M+/A+ groups dramatically increased compared with M−/A−, M+/A−, and WT groups ($P < 0.0001$, Fig. 2d). The yield of active vaccination in the M+/A+ group was $18.24 \pm 4.09$-fold the concentration of maternal Abs and remained high at 5 m ($P < 0.0001$, Fig. 2d). Exposure to maternal Abs in the M+/A− group was limited up to 1 m of age, following which Ab levels were undetectable (Fig. 2d). In actively vaccinated offspring, IgG1, IgG2a, and IgG3 levels were higher than control at 2.5 m ($P < 0.01$, Fig. 2e, f, g, respectively, summarized in Fig. 2j) and decreased to baseline at later time-points. As expected, levels of IgG2b were elevated following vaccination and remained high until 5 m of age in the M−/A+ and M+/A+ groups compared with the M−/A−, M+/A−, and WT groups ($P < 0.001$, Fig. 2h, j). Accordingly, IgM levels were elevated following vaccination in actively vaccinated mice at 2.5 m ($P < 0.0001$, Fig. 2i, j), which decreased to baseline at later time-points.

**Maternal and active vaccination rescue short-term memory abilities and normalizes exploratory behavior.** To compare the effects of maternal and active vaccinations with that of $A\beta$ on cognition, mice were tested in a battery of behavioral paradigms at 4 m of age (Fig. 1a). Short-term memory capacity was assessed using the novel object recognition (NOR) test[37]. While WT, M−/A+, and M+/A+ vaccinated 5xFAD mice exhibited a clear preference for the novel object throughout the experiment, unvaccinated mice showed no such preference after the first 20 s of the trial ($P < 0.0001$, Fig. 3a$_{1-2}$). M+/A− mice exhibited a preference to the novel object only during the first 60 s ($P < 0.01$ compared with M−/A−, $P < 0.0001$ compared with M+/A+, Fig. 3a$_{1-2}$), suggesting that maternal vaccination alone only partially ameliorated deficits in short-term memory, while active vaccination and combined vaccination resulted in a full short-term memory restoration. M−/A− mice exhibited lower exploratory behavior in this task, as they spent less time near the familiar and novel objects together, compared with M+/A− ($P < 0.01$, Fig. 3a$_2$, b). No difference was observed in the spontaneous-alternation T-maze for short-term memory assessment ($P = 0.51$, Fig. S1a). To rule out non-cognitive effects on short-term memory assessment, we also tested exploratory behavior. Similar to WT controls, M+/A+ 5xFAD mice spent more time in the center of the open field (OF) arena[38], compared with M−/A−, M+/A−, and M−/A + vaccinated 5xFAD mice ($P < 0.01$, Fig. 3c$_{1-2}$). Exploration speed and distance did not differ between groups ($P = 0.61$, $P = 0.59$, respectively, Fig. S1b-c). These results imply that a combination of maternal and active vaccination completely normalized exploratory behavior in 4m-old 5xFAD mice. To rule out anxiety-related confounds, mice were tested in the elevated

zero-maze (EZM)[39]. The fraction of time spent in the open/close zones of the maze did not differ between groups ($P = 0.96$, Fig. S1d), suggesting no effect on anxiety-related behavior. However, the number of entries to the anxiogenic open section of the EZM was lower among M−/A− mice compared with M+/A− ($P < 0.05$, Fig. 3d$_{1-2}$). Exploration distance and speed did not differ between groups ($P = 0.15$, $P = 0.15$, respectively, Fig. S1e, f). These findings suggest that while no anxiety-like behavior is found, maternal and active vaccinations enhance exploratory behavior among 5xFAD mice in the EZM.

**Maternal vaccination reduces cerebral $A\beta$ levels in adult 5xFAD mice.** Mice were sacrificed at 5 m to assess $A\beta$-related neuropathology (Fig. 1a). Cortical $A\beta_{40}$ and $A\beta_{42}$ levels were quantified using a modified sELISA protocol[40] in soluble (TBST-soluble), pre-amyloid complexes (SDS-soluble), and insoluble-fibrillary (FA-soluble) $A\beta$ extracts. M+/A− and M+/A+ mice exhibited reduced soluble $A\beta_{42}$ compared with both M−/A− and M−/A+ mice ($P < 0.05$, Fig. 4a). No difference was observed between M−/A− and M−/A+ groups ($P = 0.16$, Fig. 4a), although M−/A− was the only group to have elevated levels of soluble $A\beta_{42}$ compared with WT controls ($P < 0.05$, Fig. 4a). SDS-soluble pre-amyloid $A\beta_{42}$ was reduced in M+/A− and M−/A+ but not in M+/A+ treated mice compared with M−/A−, ($P < 0.01$, $P = 0.21$, respectively, Fig. 4b). However, M−/A− mice were the only group to have elevated levels of SDS-soluble $A\beta_{42}$ compared with WT controls ($P < 0.05$, Fig. 4b). Importantly, insoluble $A\beta_{42}$ was reduced in maternally vaccinated mice (M+/A− and M+/A+), compared with unvaccinated and actively vaccinated mice (M−/A− and M−/A+) ($P < 0.01$, Fig. 4c). No difference was observed in cortical insoluble $A\beta_{42}$ between M−/A− and M−/A+ ($P = 0.68$, Fig. 4c) or between M+/A− and M+/A+ ($P = 0.97$, Fig. 4c). Elevated levels of insoluble $A\beta_{42}$ were observed in M−/A− and M−/A+ compared with WT controls ($P < 0.01$, Fig. 4c). These findings suggest that active vaccination alone is insufficient in reducing insoluble $A\beta_{42}$. Moreover, active vaccination did not yield additional reduction in already maternally immunized mice, suggesting that early passive maternal vaccination results in a long-lasting effect on either $A\beta$ accumulation, clearance, or both.

Maternally vaccinated mice (M+/A− and M+/A+) exhibited reduced soluble $A\beta_{40}$ compared with M−/A− and M−/A+ mice ($P < 0.05$, Fig. 4d). M−/A+ mice exhibited a non-significant reduction in soluble $A\beta_{40}$ compared with M−/A− mice ($P = 0.22$, Fig. 4d). However, M−/A− was the only group in which soluble $A\beta_{40}$ was significantly higher than WT controls ($P < 0.05$, Fig. 4d). In the SDS-soluble fraction, $A\beta_{40}$ was lower in M−/A+, M+/A− and M+/A+ vaccination groups compared with unvaccinated mice ($P < 0.01$, $P < 0.01$, $P < 0.05$, respectively, Fig. 4e). Elevated levels of SDS-soluble $A\beta_{40}$ were measured in M−/A− mice compared with WT controls ($P < 0.01$, Fig. 4e). While M−/A− mice exhibited elevated levels of insoluble $A\beta_{40}$ compared with WT controls ($P < 0.01$, Fig. 4f), no difference was observed between 5xFAD groups ($P = 0.13$, Fig. 4f). Insoluble $A\beta_{42}/A\beta_{40}$ ratio[41] was significantly higher in M−/A− mice compared with WT controls ($P < 0.05$, Fig. 4g), with no difference from all other 5xFAD groups ($P = 0.13$, main effect maternal vaccination, $P = 0.16$, main effect active vaccination, Fig. 4g). To test whether the reduction in cerebral $A\beta$ among maternally vaccinated mice is a result of enhanced clearance or reduced production of $A\beta$, hAPP levels were measured at the transcript and protein levels. hAPP mRNA did not differ for both maternal ($P = 0.42$) and active ($P = 0.35$) vaccinations (Supplementary Fig. S2a), suggesting that reduced APP production cannot explain the observed reduction in $A\beta$. At the protein level, actively and

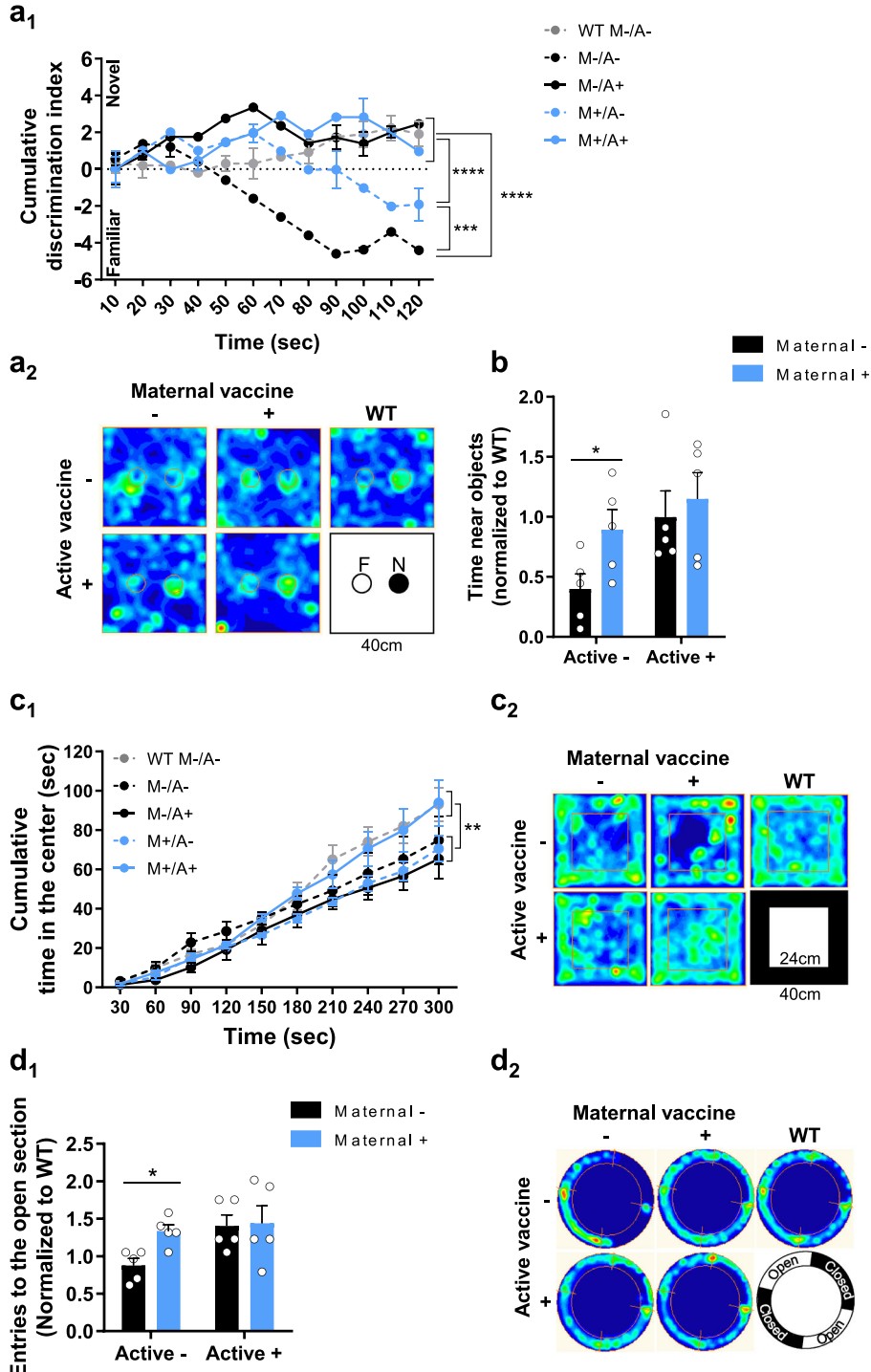

**Fig. 3 Combination of maternal and active vaccination rescues short-term memory abilities and normalizes exploratory behavior.** A behavioral battery was conducted at 4 m of age ($n = 5$) to assess the effect of maternal and active vaccinations on cognition. **a₁** Short-term memory was assessed using the NOR test. **a₂** Occupancy plots of NOR arena—F (familiar object), N (novel object). **b** Total time near objects in the NOR test. **c₁** Exploratory behavior in the OF test. **c₂** Occupancy plots of the OF arena. **d₁** Anxiety-related behavior in the EZM. **d₂** Occupancy plots of the EZM. *$P < 0.05$, **$P < 0.01$, ***$P < 0.001$, ****$P < 0.0001$, two-way ANOVA, repeated measures two-way ANOVA, data are presented as mean ± SEM.

combined vaccinated mice exhibited a reduction in hAPP compared with untreated and maternally vaccinated mice ($P < 0.05$, supplementary Fig. S2b). This effect accounts for only ~18% of the final reduction seen in cerebral Aβ levels and is possibly due to the neutralization of hAPP by antibodies. These results suggest that most of the observed reduction in Aβ is not due to lowered hAPP expression and is, therefore, attributable to Aβ clearance. Maternal anti-Aβ IgG levels in offspring circulation negatively correlated with insoluble Aβ₄₂ levels at 5 m of age ($r = -0.43$, $P < 0.05$, Fig. 4h, S3a), whereas active vaccination showed no such correlation ($r = -0.11$, $P = 0.32$, Fig. 4h, S3d). Similar correlations were found for maternal IgG2b and IgM levels

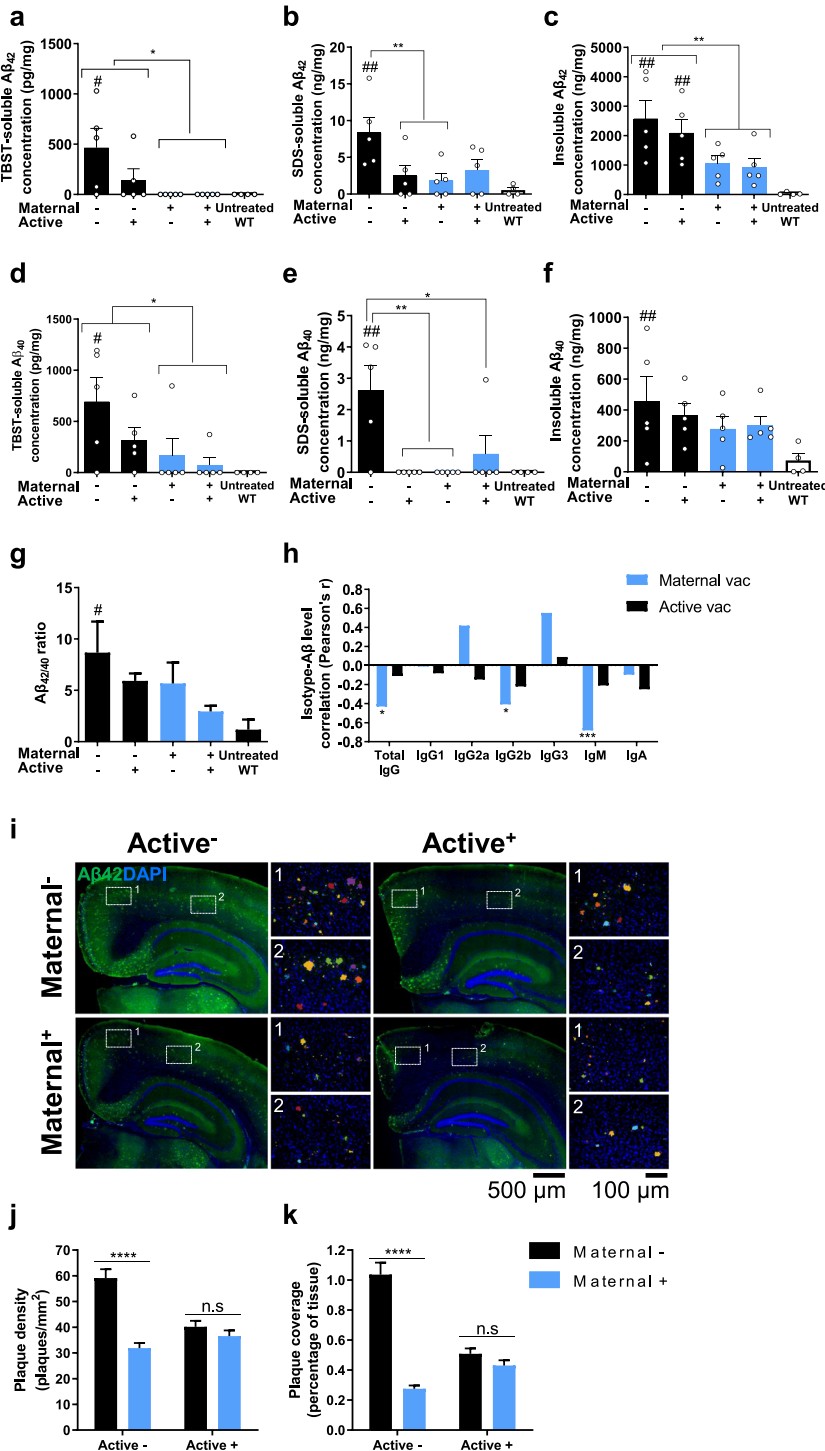

**Fig. 4 Maternal vaccination reduces Aβ pathology independently of anti-Aβ antibody presence at adulthood.** Cortical Aβ levels were measured using sELISA (**a**) TBST, (**b**) SDS, and (**c**) formic-acid soluble Aβ$_{42}$ ($n = 5$). (**d**) TBST, (**e**) SDS, and (**f**) formic-acid soluble levels of cortical Aβ$_{40}$ ($n = 5$). **g** Cortical Aβ42/40 ratio. **h** Insoluble Aβ levels at 5 m of age negatively correlate with IgG2b and IgM levels, following maternal vaccination. **i** Plaque load was quantified using immunofluorescence and blob detection. Plaques were pseudocolored in high magnification images to better separate and emphasize the different plaques. **j** Quantification of plaque density (plaques/mm$^2$) and (**k**) Plaque tissue coverage (percentage), *$P < 0.05$, **$P < 0.01$, ****$P < 0.0001$, #$P < 0.05$, ##$P < 0.01$ compared to WT controls, two-way ANOVA, one-way ANOVA, Pearson's correlation, data are presented as mean ± SEM.

($r = -0.4$, $r = -0.68$, respectively, $P < 0.05$, $P < 0.001$, respectively, Fig. 4h, S3b, S3c), while active-vaccination-derived IgG2b and IgM did not correlate with a reduction in Aβ load ($r = -0.21$, $r = -0.1$, respectively, $P = 0.17$, $P = 0.18$, respectively, Fig. 4h, S3e, S3f). M+/A− exhibited reduced plaque density in the cortex

compared with M−/A− ($P < 0.0001$, Fig. 4i, j). Accordingly, the area covered by plaques was also reduced in M+/A− compared with M−/A− mice ($P < 0.0001$, Fig. 4I, k). Active vaccination had a minor contribution to plaque density reduction in already maternally vaccinated mice ($P = 0.49$, Fig. 4i, j).

Altogether, these findings provide evidence for a long-lasting effect of maternal vaccination during developmental and post-natal periods on Aβ clearance or accumulation, with a limited contribution of active vaccination at early adulthood.

**Maternal vaccination induces long-term FcRs upregulation in the brain**. A limited-time exposure of offspring to MAbs resulted in a long-lasting effect on Aβ pathology and short-term memory restoration, months after MAbs were not detected in the circulation. As Aβ was not targeted by MAbs later than 1 m of age, two mechanisms could be ruled out: (a) maternal Abs interfere with Aβ oligomerization and fibrillation, to prevent plaque formation[42], and (b) maternal vaccination facilitates Aβ clearance by microglia via opsonization[43]. This led us to hypothesize that maternal vaccination promoted a long-lasting microglial phagocytic phenotype, independently of the presence of anti-Aβ Abs. To verify this, we assessed the expression of various Fc receptors in the brain at 1 m following maternal vaccination only, and at 5 m of age, following maternal and active vaccination. The activating FcγRI, FcγRIII, and FcγRIV initiate Fc-mediated phagocytosis via several actin-regulating pathways[44,45]. Additionally, FcRn is crucial for the transport of maternal Abs from the placenta and gut to the circulation of fetuses and newborns[46]. At 1 m, cerebral FcγRI levels were elevated in maternally vaccinated WT and 5xFAD mice compared with sham-vaccinated controls ($P < 0.05$, $P < 0.001$, respectively, Fig. 5a). Interestingly, FcγRIII levels increased in maternally vaccinated 5xFAD mice compared with controls ($P < 0.001$, Fig. 5b), but not among vaccinated WT mice compared with WT controls ($P = 0.12$, Fig. 5b). A similar strain-specific effect was also observed for FcγRIV expression in the brain ($P = 0.07$ for WT, $P < 0.0001$ for 5xFAD, Fig. 5c). FcRn levels were elevated in both WT and 5xFAD mice compared with controls ($P < 0.01$ for WT, $P < 0.01$ for 5xFAD, Fig. 5d). Interestingly, levels of the inhibitory FcγRIIb were elevated in vaccinated 5xFAD mice compared with controls ($P < 0.01$, Fig. 5e), but not in vaccinated WT mice ($P = 0.18$, Fig. 5e). These results indicate that the presence of MAbs correlates with transcript upregulation of their receptors in the brain. At 5 m, FcγRI levels in M+/A−, M−/A+, and M+/A+ mice were elevated compared with M−/A− mice ($P < 0.05$, Fig. 5f). No difference was observed between M+/A− and M+/A+ mice, implying the absence of additional contribution of active vaccination to FcγRI upregulation in already maternally vaccinated mice. FcγRIII mRNA levels were higher in M+/A− and M−/A+ mice compared with unvaccinated mice ($P < 0.05$, Fig. 5g). However, combined maternal and active vaccination M+/A+ did not differ significantly from unvaccinated mice ($P = 0.07$, Fig. 5g). FcγRIV transcript levels did not differ between groups ($P = 0.27$, the main effect for maternal vaccination, $P = 0.47$, main effect for active vaccination, Fig. 5h). FcRn levels in M+/A−, M+/A+, and M+/A+ mice were elevated compared with M−/A− mice ($P < 0.05$, Fig. 5i). Levels of inhibitory FcγRIIb were elevated in M+/A− and M−/A+ compared with M−/A− ($P < 0.05$, $P < 0.01$, respectively, Fig. 5j). Importantly, FcγRIIb levels were reduced in M+/A+ compared with M−/A+ mice ($P < 0.05$, Fig. 5j), suggesting that the combined vaccination results in upregulation of activator FcRs and downregulation of inhibitory FcRs. In addition, elevation in FcγRI, FcγRIII, and FcRn strongly correlated with Aβ reduction (full data are described in the supplementary note, Fig. S4a–c, e).

Triggering receptor expressed on myeloid cells-2 (TREM2), an innate immune receptor expressed on microglia, is heavily implicated in microglial recruitment to Aβ plaques[47–49]. DNAX-activating protein of 12 kDa (DAP12; also known as TYROBP)

associates with TREM2 and is essential for its function in signal transduction following ligand binding[50,51]. Intriguingly, 1m-old maternally vaccinated WT and 5xFAD mice exhibited high expression of TYROBP compared with controls ($P < 0.01$, Supplementary Fig. S4g). TYROBP expression remained elevated months after maternal vaccination, as its expression in M+/A− mice was higher compared with M−/A− mice ($P < 0.05$, Supplementary Fig. S4h). M−/A+ and M+/A+ exhibited a non-significant elevation in TYROBP expression at 5 m of age compared with controls (Supplementary Fig. S4h). This suggests that in addition to upregulating FcγR, maternal vaccination also enhances the expression of TYROBP in a long-lasting manner, which supports TREM2 signaling in microglia.

**Maternal vaccination elevates FcγRI on microglial cells**. As FcγR signaling could mediate the effects of maternal vaccination on Aβ pathology, we next assessed FcγRI, FcγRIIb, FcγRIII, and FcγRIV expression in CNS cells. FcγRI was abundantly expressed in microglia, but not in neurons or astrocytes (full data are described in the supplementary note, Figs. S5–6). Consequently, we assessed FcγRI levels in microglial cells in the hippocampus and cortex of maternally and actively immunized mice. At 5 m, hippocampal FcγRI levels were elevated in M+/A− and M+/A+ microglia compared with both M−/A− and M−/A+ ($P < 0.05$, $P < 0.01$, respectively, Fig. 6a-b). Active vaccination alone did not yield such elevation compared with M−/A−($P = 0.84$, Fig. 6a, b). FcγRI expression was normally distributed in M+/A+ microglia ($P = 0.7$ compared with the simulated normal distribution of the same mean and standard deviation, Fig. 6c) and close to the normal distribution in M+/A− ($P = 0.04$, Fig. 6c). In M−/A− and M−/A+, FcγRI distribution was skewed to the right and significantly differed from the simulated normal distribution ($P < 0.0001$, Fig. 6c). FcγRI distribution in M+/A− and M+/A+ hippocampal microglia differed from M−/A− and M−/A+ ($P < 0.0001$, Fig. 6d), with no difference between M+/A− and M+/A+ mice, suggesting no additive effect of active vaccination on already maternally immunized animals. Similar effects were found in cortical microglia, as FcγRI levels were elevated in M+/A− and M+/A+ mice, compared with controls (full data are described in the supplementary note, Fig. S7).

**Maternal vaccination activates phagocytosis-related signaling cascades in microglial cells**. Maternal vaccination upregulated FcγRI, FcγRIII, FcγRIV, and FcRn transcripts (Fig. 5a–j) and increased microglial FcγRI protein expression (Fig. 6a–d). Clustering of FcγRs induces activation of Src-related kinases that phosphorylate the immunoreceptor tyrosine-based activation motif (ITAM) domain on FcγRs. Syk is recruited to the receptor and undergoes autophosphorylation. This in turn activates downstream signaling molecules, such as AKT, ERK, and cofilin, all involved in actin-cytoskeleton regulation and are essential for phagocytosis[52].

At 5 m of age, pSyk was upregulated in the hippocampus of M+/A− and M+/A+ compared with M−/A− and M−/A+ mice ($P < 0.05$, Fig. 7a, b). Similarly, total Syk was also upregulated in maternally vaccinated mice ($P < 0.001$, Fig. 7a-b), suggesting that FcγR upregulation by maternal vaccination resulted in activation of downstream Syk signaling. pAKT levels were elevated as a result of active vaccination in M−/A+ and M+/A+ compared with M−/A− and M+/A−, with no effect of maternal vaccination ($P < 0.01$, Fig. 7a, c). Total AKT levels did not differ between groups ($P = 0.9$, main effect for active vaccination, 0.76, main effect for maternal vaccination, Fig. 7a, c). In a similar manner, pERK levels increased in actively vaccinated mice M−/A+ and M+/A+ compared with

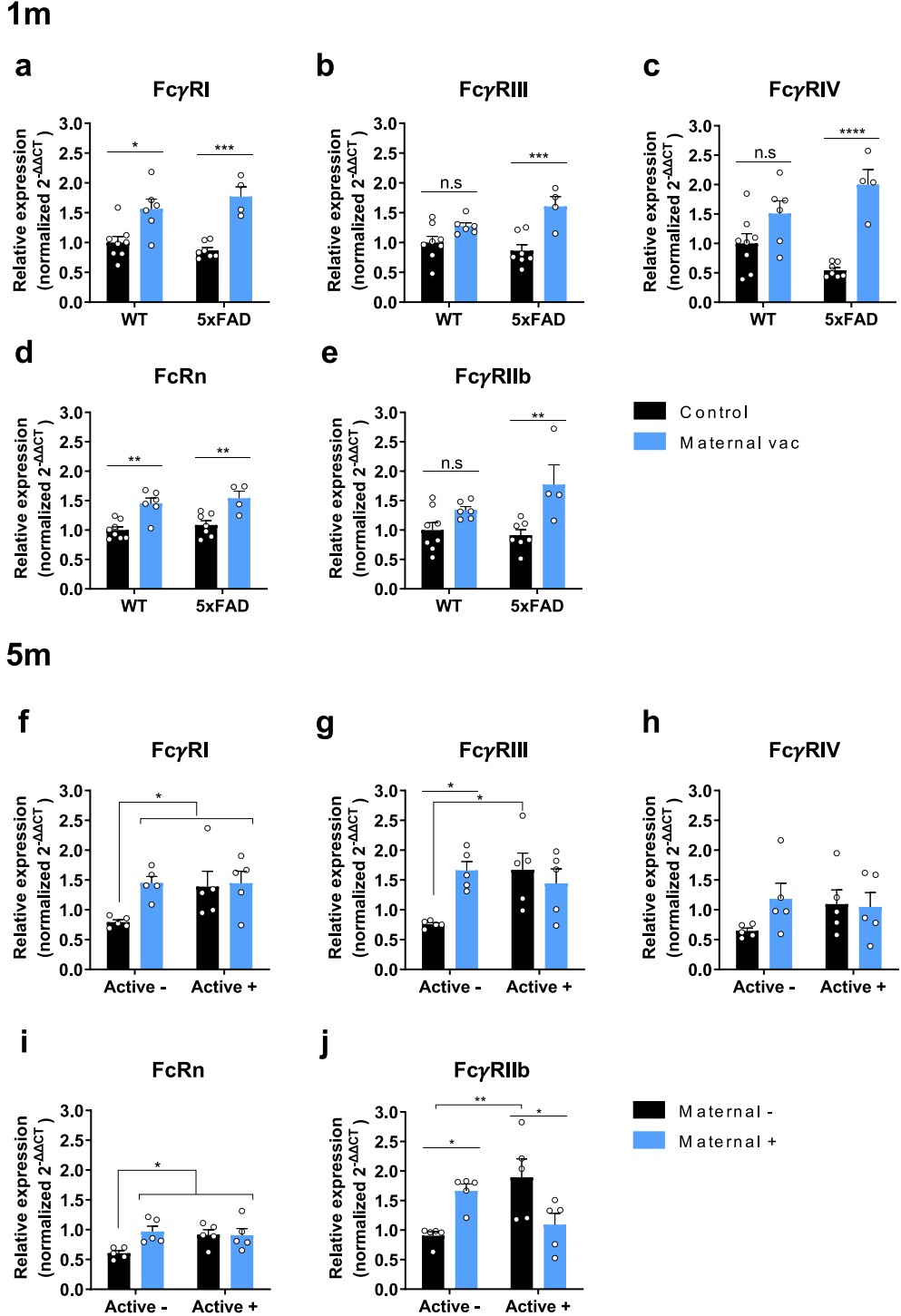

**Fig. 5 Maternal vaccination induces long-term FcRs upregulation in the brain.** Levels of cerebral FcR mRNA were quantified using RT-qPCR after maternal vaccination at 1 m of age (**a–e**) and following maternal and active vaccination at 5 m of age (**f–j**). mRNA levels of (**a**) FcγRI, (**b**) FcγRIII, (**c**) FcγRIV, (**d**) FcRn, and (**e**) FcγRIIb in 1m-old WT (n = 14) and 5xFAD (n = 11) mice. Levels of (**f**) FcγRI, (**g**) FcγRIII, (**h**) FcγRIV, (**i**) FcRn, and (**j**) FcγRIIb in maternally and actively vaccinated 5m-old 5xFAD mice (n = 5). *$P < 0.05$, **$P < 0.01$, ***$P < 0.001$, ****$P < 0.0001$, two-way ANOVA, data are presented as mean ± SEM.

M−/A− and M+/A−, with little contribution of maternal vaccination ($P < 0.05$, Fig. 7a, d). Total ERK levels did not differ between groups ($P = 0.98$, main effect for active vaccination, 0.54, main effect for maternal vaccination, Fig. 7a, d). This may serve as evidence for Syk activation of ERK and AKT-mediated actin

regulation as a result of active, but not maternal, vaccination. pCofilin levels increased in maternally immunized mice M+/A−, compared with unvaccinated mice M−/A− ($P < 0.05$, Fig. 7a, e). No difference was observed between actively M−/A+ and combined M+/A+ vaccination ($P = 0.87$, Fig. 7a, e). Moreover,

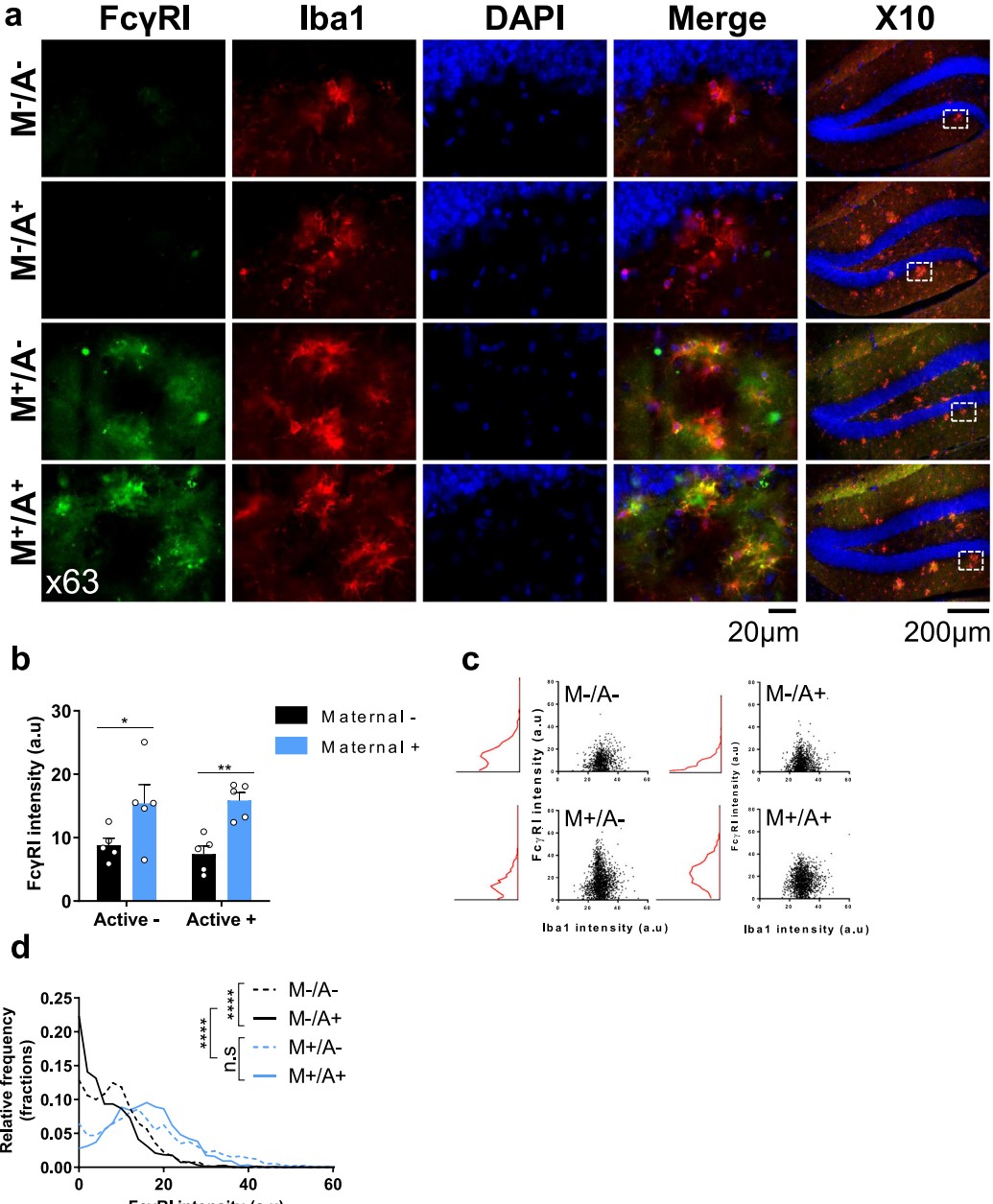

**Fig. 6 Maternal vaccination upregulates microglial FcRs expression.** Microglial FcγRI protein expression was quantified using immunofluorescence following maternal and active vaccination at 5 m of age ($n = 5$). **a** FcγRI expression was assessed using double-labeled immunofluorescence in Iba1+ microglia, using the x10 and x63 objectives for visualizing and quantification, respectively. **b** Quantification of FcγRI signal intensity among microglia. **c** Scatter plot and intensity distribution FcγRI signals in Iba1+ cells. **d** Overlay and comparisons of FcγRI expression distribution. *$P < 0.05$, **$P < 0.01$, ****$P < 0.0001$, two-way ANOVA, corrected two-sample Kolmogorov–Smirnov test, data are presented as mean ± SEM.

maternal vaccination alone yielded similar pCofilin levels as did active vaccination or combined vaccine. Levels of cofilin were reduced in actively M−/A+ and combined M+/A+ mice compared with unvaccinated M−/A− and maternally vaccinated M−/A+ mice ($P < 0.05$, Fig. 7a, e). This may serve as evidence that maternal vaccination enhances actin polymerization via Syk-mediated elevation in pCofilin levels and reduction in Cofilin levels. Finally, we found that both maternal, M+/A− and M+/A+, and active M −/A+ vaccination reduce actin levels compared with unvaccinated M−/A− mice ($P < 0.05$, Fig. 7a, f). These findings provide evidence for maternal vaccination's long-term effect in facilitating FcR-mediated phagocytosis signaling pathways (Fig. 7g).

**Maternal vaccination facilitates long-term Aβ clearance via enhancing FcR-mediated microglial phagocytosis.** FcR upregulation led to the recruitment and activation of Syk, which in turn initiated an actin-regulating pathway via AKT, ERK, and Cofilin. Regulation of actin polymerization and depolymerization is essential for the construction of the phagosome[44]. CD68, a lysosomal protein expressed in macrophages and activated microglia, is often associated with pro-inflammatory disease-associated microglia. To assess whether maternally and actively vaccinated mice exhibit elevated phagocytic behavior, immunofluorescence (IF) analysis of CD68 among hippocampal Iba1+ cells was conducted. Maternally vaccinated mice M+/A− exhibit elevated CD68 levels in Iba1+ cells, compared with M−/A−

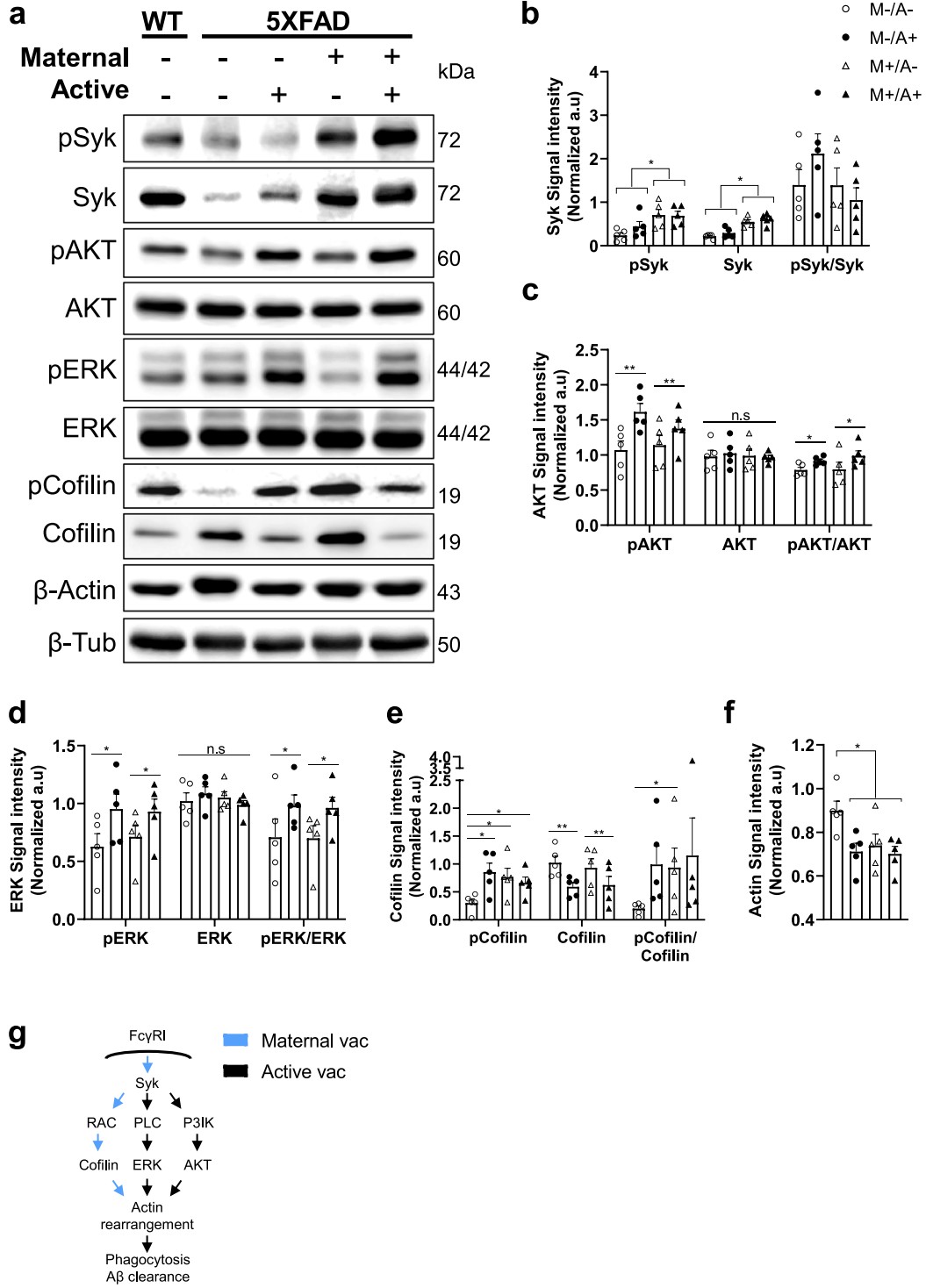

**Fig. 7 Maternal vaccination activates actin-cytoskeleton regulation pathways via Syk, AKT, ERK, and Cofilin phosphorylation.** Level of Syk and downstream signaling molecules from the Fc-mediated phagocytosis pathway were measured using immunoblotting following maternal and active vaccination at 5 m of age ($n = 5$). Levels of (**a**, **b**) pSyk/Syk, (**a**, **c**) pAKT/AKT, (**a**, **d**) pERK/ERK, (**a**, **e**) pCofilin/Cofilin, and (**a**, **f**) Actin levels, normalized to tubulin. **g** Maternal and active vaccinations activate FcγR-mediated phagocytosis pathways through actin cytoskeleton regulation. *$P < 0.05$, **$P < 0.01$, two-way ANOVA, data are presented as mean ± SEM.

($P < 0.05$, Fig. 8a-b). Moreover, M−/A+ and M+/A+ mice exhibit higher CD68 expression than M−/A− and M+/A− mice ($P < 0.0001$, Fig. 8a, b), with no difference between M−/A+ and M+/A+ ($P = 0.75$, Fig. 8a, b). These findings suggest that maternal vaccination alone enhances microglial phagocytosis. Ramified microglia were also observed in the hippocampi of naïve, maternally, and actively vaccinated mice (Supplementary

Fig. S8a). At the transcript level, M+/A− mice exhibit elevated levels of *CD68* mRNA compared with M−/A− ($P < 0.001$, Fig. 8c), suggesting that maternal vaccination support microglial phagocytosis even in the absence of MAbs, possibly via FcR-Syk activation. Additionally, actively vaccinated M−/A+ and combined M+/A+ vaccination mice also exhibit elevated transcript levels of *CD68* compared with unvaccinated controls ($P < 0.05$,

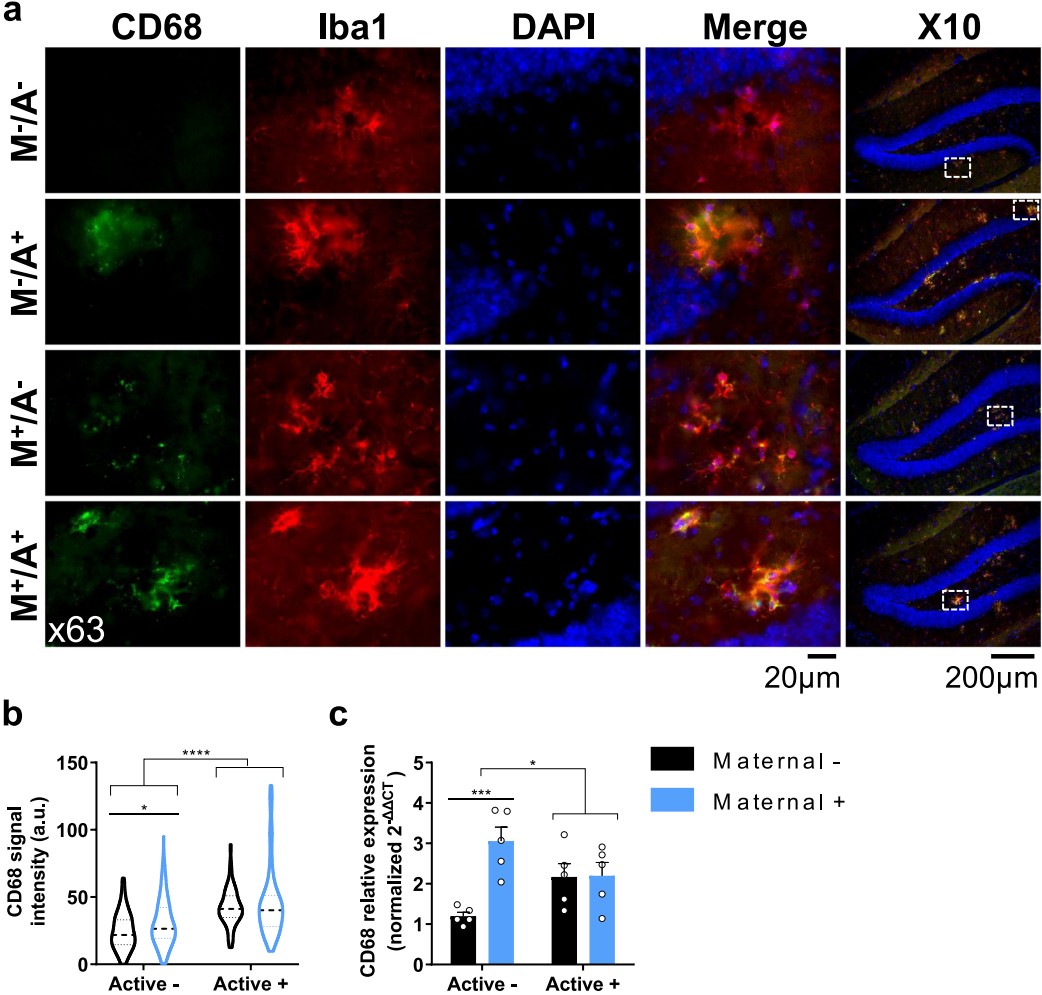

**Fig. 8 Maternal vaccination enhances microglial phagocytosis.** Microglia reactivity in the hippocampus was assessed using double labeling of CD68 and Iba1+ microglia following maternal and active vaccination at 5 m of age ($n = 5$). **a** Double labeling of CD68 and Iba1+ microglia using the x10 and x63 objectives for visualizing and quantification, respectively. **b** Quantification of CD68 signal intensity, lines represent the median and the 25, 75 percentiles. **c** CD68 mRNA levels were measured using RT-qPCR. *$P < 0.05$, ***$P < 0.001$, ****$P < 0.0001$, two-way ANOVA, data are presented as mean ± SEM.

Fig. 8c). These data imply that microglial phagocytosis is enhanced in maternally and actively vaccinated mice. To support the maternal vaccination/FcR/Syk/phagocytosis hypothesis, we stained hippocampal CD68+ cells for pSyk. Maternally vaccinated M+/A− mice exhibited elevated pSyk levels among CD68+ cells in the hippocampus, compared with M−/A− mice ($P < 0.05$, Fig. S9a, b), suggesting that maternal vaccination alone is sufficient in inducing Syk activation in microglial cells, long after maternally induced Abs are abolished. Additionally, maternally and actively M+/A+ mice exhibited elevated microglial pSyk levels compared with actively M−/A+ immunized mice ($P < 0.0001$, Fig. S9a, b), suggesting that active vaccination alone does not induce Syk activation as strongly as maternal vaccination.

**MAbs facilitate Aβ clearance by microglial cells in a Syk-dependent manner.** We next assessed the effect of MAbs on Syk-dependent phagocytic capacity of N9 murine embryonic microglial cell-line[53] and adult primary microglia. Increasing concentration (0.75–5 μM) of BAY-61-3606 (BAY), a highly selective Syk inhibitor[54] in the media of N9 cells reduced Aβ-induced Syk phosphorylation in a dose-dependent manner (Fig. 9a). Accordingly, alteration in downstream signaling molecules of the Fc-mediate phagocytosis pathway was observed,

namely, a reduction in ERK and AKT phosphorylation and an increase in cofilin phosphorylation (Supplementary Fig. S10a, c). As activation of ERK and AKT exhibited a considerable reduction in cells incubated with 5 μM of BAY, the following experiments were conducted using this concentration. We first assessed the general capacity of N9 cells to perform phagocytosis of FBS-coated fluorescent beads following a 3 h FBS deprivation. N9 cells treated with BAY exhibited reduced phagocytic activity compared with controls ($P < 0.001$, Fig. 9b, c) and a reduction in FITC intensity ($P < 0.05$, Fig. 9d), reflecting not only a lower percentage of phagocytic cells but also a lower number of intracellular beads. Importantly, Syk inhibition reduced phagocytosis of Aβ42 peptide, as the number of Aβ foci per mm² was reduced by ~50% following BAY administration compared to naïve cells ($P < 0.0001$, Fig. 9e, f). Next, we tested whether pre-incubation of Aβ with serum from anti-Aβ vaccinated dams facilitates phagocytosis of Ab–Ag complexes. N9 cells were treated with Aβ42 alone, Aβ42 pre-incubated with serum from sham, or anti-Aβ vaccinated dams with and without BAY. 5.1% of untreated N9 cells were Aβ positive, demonstrating baseline phagocytosis capacity of these cells (Fig. 9g, h). Incubation with serum from sham-vaccinated resulted in 11.7 ± 0.03% of phagocytic cells ($P < 0.0001$ compared to baseline, Fig. 9g, h), and incubation with serum from anti-Aβ vaccinated dams elicited phagocytosis in 81 ± 0.003% of the cells

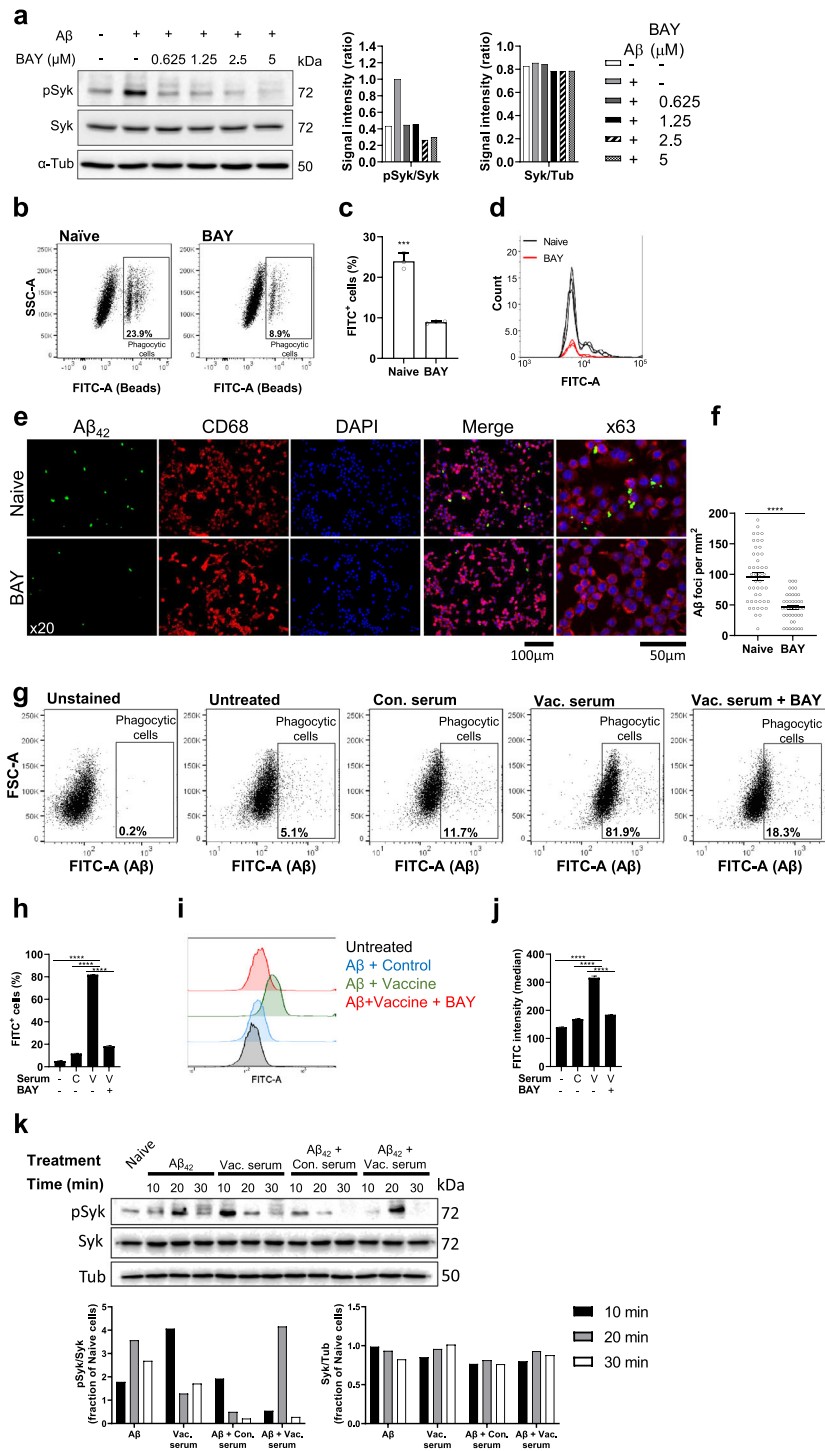

**Fig. 9 Maternal antibodies mediate Aβ phagocytosis by N9 microglial cell-line in a Syk-dependent manner.** Phagocytosis capacity and intracellular signaling were assessed in N9 murine embryonic cell-line following treatment with either Aβ42 alone or Aβ42 pre-incubated with serum from anti-Aβ or sham-vaccinated dams. The contribution of Syk signaling was assessed using Syk inhibition by BAY-61-3606. **a** BAY inhibits Syk phosphorylation in a dose-dependent manner. **b–d** Phagocytosis of FBS-coated fluorescent beads by N9 cells is reduced following treatment with Syk inhibitor. **e, f** IF of N9 cells following incubation with Aβ reveals a reduction in phagocytosis capacity after Syk inhibition. **g–j** Serum from anti-Aβ vaccinated dams increases phagocytosis by N9 cells compared with serum from sham-vaccinated dams and abrogation of this effect following Syk inhibition. **k** Treatment with Aβ, serum from anti-Aβ or sham-vaccinated dams, or combined treatment reveals differences in the time-course of Syk phosphorylation. ***$P < 0.001$, ****$P < 0.0001$, unpaired t-test, Chi-squared test for independence, Fisher's exact test, one-way ANOVA, data are presented as mean ± SEM.

($P < 0.0001$ compared with baseline and sham-vaccinated serum, Fig. 9g, h). Importantly, this drastic effect was abrogated by Syk inhibition, as only $18.3 \pm 0.003\%$ of BAY and serum-treated cells were phagocytic ($P < 0.0001$, Fig. 9g, h). The median of FITC intensity increased among cells treated with anti-Aβ serum compared with sham-vaccinated serum ($P < 0.0001$, Fig. 9i, j) and reduced following treatment with BAY ($P < 0.0001$, Fig. 9i, j). These results demonstrate that MAbs facilitate Fc-receptor mediated Aβ phagocytosis in a Syk-dependent pathway. To further understand the alteration in cellular signaling following serum and Aβ administration, Syk phosphorylation was measured using western blotting 10–30 min after treatment with either $Aβ_{42}$ alone, serum from anti-Aβ vaccinated dams, or $Aβ_{42}$ pre-incubated with serum from vaccinated and sham-vaccinated dams (Fig. 9k). Aβ alone elicited elevation in Syk activation (3.75-fold change of naïve cells after 20 min, Fig. 9k). Serum from anti-Aβ vaccinated dams alone also resulted in Syk activation (4.06-fold change of naïve cells after 10 min, Fig. 9k). The difference in Syk activation timing implies that Aβ and antibodies alone may activate different signaling pathways, e.g. Fc-receptor signaling by antibodies and TREM2 signaling by Aβ, both of which are Syk-dependent. Interestingly, serum from sham-vaccinated dams elicited a milder activation of Syk (1.92-fold change of naïve cells after 10 min, Fig. 9k), supporting our hypothesis that Syk phosphorylation associated with Fc receptors peaks 10 min following treatment. Treatment with both Aβ and serum from vaccinated dams yielded a strong but transient Syk activation (4.16-fold change of naïve cells after 20 min, Fig. 9k), suggesting that Ab–Ag complexes generate a somewhat different effect than Ab or Ag alone.

To further examine the contribution of MAbs to Fc-receptor Syk-mediated Aβ clearance by microglia, a phagocytosis assay was conducted in adult primary microglia from naïve WT mice. Cells were administered with $Aβ_{42}$ (750 nM) following incubation with serum from sham-vaccinated dams, anti-Aβ vaccinated dams, and Syk inhibitor BAY. 4 h following Aβ administration, primary microglial cells were fixed and stained for CD68 and intracellular Aβ. Treatment with serum from sham-vaccinated females did not increase Aβ phagocytosis compared with untreated cells ($P = 0.37$, Fig. 10a, b). In contrast, treatment with anti-Aβ serum increased phagocytic activity compared with both untreated and sham-vaccinated serum treatment ($P < 0.001$, Fig. 10a, b). Importantly, this effect was abrogated by Syk inhibition ($P < 0.01$, Fig. 10a, b), demonstrating that Aβ clearance facilitated by antibodies is a Syk-dependent process.

## Discussion

To date, there is no effective therapy for AD nor AD-related neuropathology and dementia in DS. Considering the positive outcomes of active vaccination in 3xTg-AD and Ts65Dn mouse models of fEOAD and DS[27,29] and the rapid propagation of cerebral Aβ in these conditions, maternal vaccination may serve as an early, preventative intervention, well before the formation of Aβ plaques. So far, anti-Aβ vaccination targeting dementia in AD patients has failed to translate to the clinic. However, the age of intervention in relation to the onset of the disease and the subjects' immunological age might be held responsible[55–57]. The DS case is, therefore, quite different in this manner. DS is detectable in utero, and the progression of Aβ pathology is earlier and faster in DS than in late-onset AD. We thus propose that early anti-Aβ intervention in the form of maternally transferred Abs, along with active postnatal vaccination, may provide continuous immune targeting of Aβ in individuals with DS. Moreover, should prenatal testing for fEOAD-related mutations eventually translate to the clinic as a diagnostic tool, the maternal vaccination strategy may

also benefit individuals prone to early Aβ deposition, such as individuals carrying the L166P mutation in PSEN1, who exhibit disease onset at adolescence[11].

In this study, we vaccinated WT females prior to their pregnancy with transgenic embryos, as the duration of mature IgG production exceeds the length of pregnancy in mice. In humans, full humoral response can be achieved during pregnancy. In contrast to the full $Aβ_{42}$, using the 1-11 fragment of Aβ, which does not contain cytotoxic T-cell epitopes, reduces the risk of a hazardous response[27]. Moreover, WT mice that received anti-$Aβ_{1-11}$ vaccine did not exhibit any behavioral or cognitive decline, neuronal loss, or inflammatory response compared with controls[29]. To date, more than 50 clinical trials have been conducted using DNA delivery by electroporation, reporting enhancement of DNA immunogenicity in humans and tolerability by experimental animal and human subjects without causing substantial negative side effects[58,59].

The transfer of IgG2b Abs from dams to pups is supported by elevation of FcRn expression, which facilitates the passage of maternal Abs through the placenta and the crossing of maternal Abs from the gut of newborns into the circulation[46]. Moreover, FcRn plays a role in elongating IgG half-live as it restores opsonized IgG and prevents its degradation. Maternal vaccination thus initiates a positive regulation loop in which the presence of MAbs elevates the transcription of FcRn, which in turn supports the passage of these Abs through the placenta and lactation and elongates their half-live and functionality. Among the MAbs milieu, IgG2b was the most common isotype that crossed the placenta. IgG2b and IgM were the most common isotypes delivered through lactation. Since IgM cannot cross the blood-brain barrier (BBB)[60], IgG2b is the main candidate effector for any downstream immunomodulation inside the CNS. Accordingly, we found elevated levels of FcγRI and FcγRIII, which bind IgG2b[45], but no expression of cerebral $Fcα/μR$ that binds IgA and IgM. The positive correlation between maternal and newborns IgG2b levels along with the negative correlation between IgG2b levels in newborns and their Aβ levels at 5 m strongly support this suggestion. Indeed, we found that high maternal IgM levels in newborns were correlated with low Aβ pathology at 5 m. The contribution of IgM, if exists, is thus attributed to peripheral rather than CNS immunomodulation. Following active vaccination, IgG2b was the only isotype to remain high three months after the last boost. Taken together, IgG2b remains the main effector of both maternal and active vaccination.

We hypothesized that a combination of maternal and active vaccination would yield continuous protection from Aβ neurotoxicity. Indeed, this combination resulted in normalization of exploratory behavior and restoration of short-term memory capacity. Intriguingly, maternal vaccination alone had no effect on exploratory behavior and only a partial positive effect on short-term memory. In contrast, active vaccination yielded a full rescue of short-term memory ability. Reduction in Aβ levels, however, was strongly dependent on maternal vaccination. Active vaccination produced a minor reduction in insoluble Aβ levels, whereas maternal vaccination, with or without the addition of active vaccination, greatly reduced levels of this fraction. Moreover, MAbs concentration in the newborns' serum was significantly more predictive of Aβ clearance than actively induced-Abs in young adult mice. Taken together, the cognitive and neuropathological findings indicate that the combination of maternal and active vaccination produced the most potent therapeutic effect.

MAbs were present in the offspring's circulating blood for the short and limited period of pregnancy and lactation. Following weaning, these Abs were no longer detectable. However, four months after Abs were absent, Aβ levels were dramatically

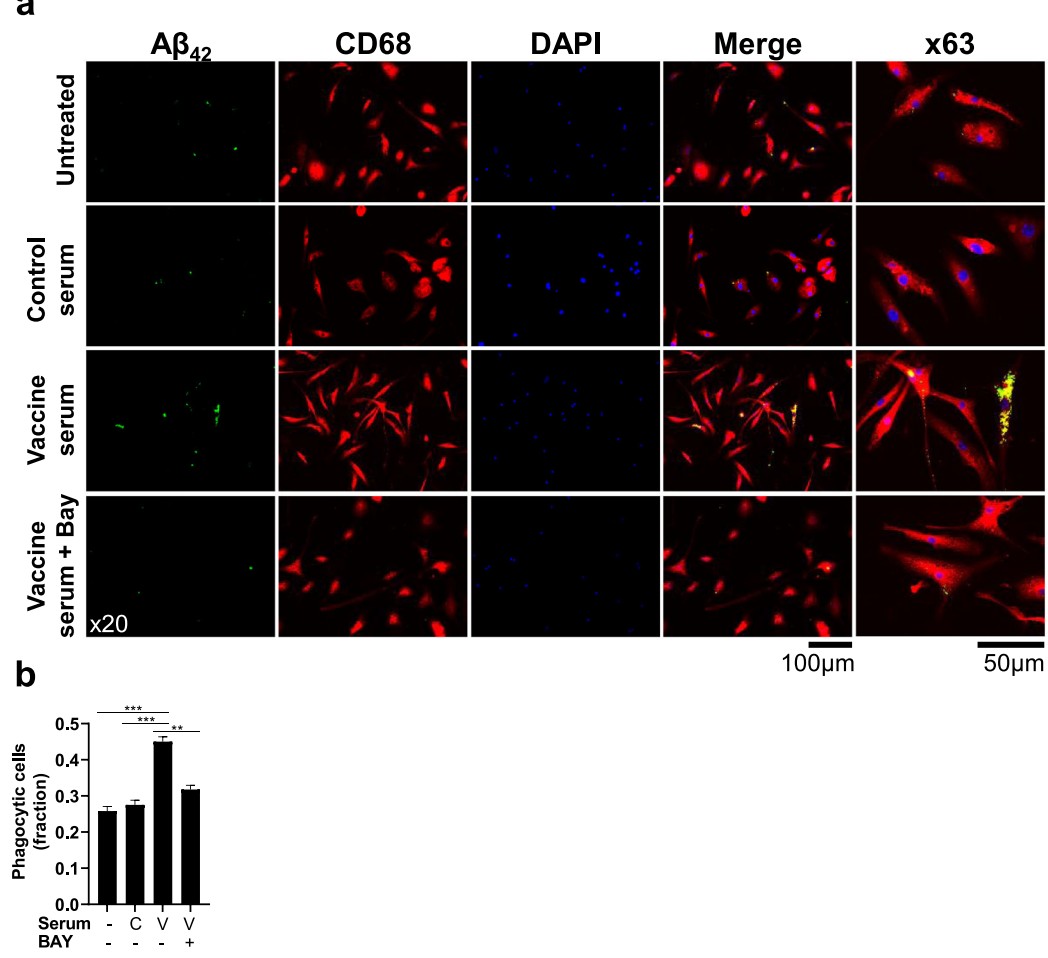

**Fig. 10 Maternal antibodies mediate Aβ phagocytosis by primary adult microglia in a Syk-dependent manner.** Aβ phagocytosis capacity of adult primary microglia was assessed following treatment with either Aβ$_{42}$ alone or Aβ$_{42}$ pre-incubated with serum from anti-Aβ or sham-vaccinated dams. The contribution of Syk signaling was assessed using Syk inhibition by BAY-61-3606. **a** Immunofluorescence images and (**b**) quantification of Aβ phagocytosis by adult primary microglia reveal that serum from anti-Aβ vaccinated dams facilitates Aβ clearance in a Syk-dependent pathway. $**P < 0.01$, $***P < 0.001$, Chi-squared test for independence, Fisher's exact test, data are presented as mean ± SEM.

reduced, implying a long-lasting immunomodulatory effect on microglial cells. Maternal vaccination indeed increased the expression of effector FcγRs and FcRn in the brain. Notably, the expression levels of Fc receptors negatively correlated with insoluble Aβ levels. Fc receptors, expressed on various immune cells, constitute critical elements for activating or downregulating immune responses[61]. Our data suggest that an increase in FcR ligands upregulates the expression of FcγRI, FcγRIII, and FcγRIV at the transcript level in the brain, and FcγRI at the protein level in microglial cells. The effector FcγRI contains an intracellular immunoreceptor tyrosine-based activation motif (ITAM) domain, which mediates cellular activation triggered by Ab-receptor binding[61]. Clustering of FcγRs induces activation of Src family kinases that phosphorylates the ITAM domain within the receptor. When an effector FcR is activated, Syk is recruited to the receptor and undergoes autophosphorylation, resulting in activation of downstream signaling molecules, such as AKT, ERK, and cofilin, all involved in actin-cytoskeleton regulation and are essential for phagocytosis[52]. The FcγRs-Syk pathway is heavily implicated in phagocytosis[62,63], a central function of microglial cells in the context of targeting amyloid burden[35]. Indeed, we found that Syk was activated in maternally vaccinated mice. Accordingly, levels of phospho-cofilin were increased, and levels of cofilin decreased. The actin-depolymerizing factor (ADF)/

cofilin protein family consists of small actin-binding proteins that play central roles in accelerating actin turnover by disassembling actin filaments[64]. The activity of cofilin is regulated by phosphorylation on residue Ser3 by LIM kinases (LIMK1 and LIMK2) and TES kinases (TESK1 and TESK2), which inhibit its interaction with actin[65]. Thus, maternal vaccination-induced FcR-Syk mediated cofilin phosphorylation contributes to the inactivation of cofilin and promotes actin polymerization, which is essential for phagosome formation in microglia[44]. ERK and AKT activation is also implicated in Fc-mediated phagocytosis, as both molecules undergo phosphorylation under Syk activation. Inhibition of Src and Syk kinases suppresses phagocytosis in primary human microglia, suppressing ERK and AKT activity[62]. Following the onset of Aβ propagation, ERK and AKT activation increased in actively, but not maternally, vaccinated mice.

CD68 is a heavily glycosylated type I transmembrane glycoprotein, mainly associated with endosomal/lysosomal compartments[66]. In the brain, it is highly expressed in phagocytic microglia[67]. In accordance with previous results, maternally vaccinated mice express higher levels of microglial CD68, suggesting enhanced phagocytic activity. In addition to FcγR-Syk axis activation in microglia, we found that maternal vaccination upregulates the expression of *TYROBP*, a signaling adapter protein essential for TREM2 signaling in microglia[50,51]. We speculate

that TREM2-mediated microglial activation is a secondary and complementary event, initiated following FcγR-upregulation, as FcRs, rather than TREM2, are the primary receptor family for IgG antibodies. Moreover, TREM2 is required for the full phagocytic capacity of antibody–Aβ complexes (a process mediated by FcR for phagocytosis) by the N9 mouse microglial cell line and primary microglia, showing that TREM2 enables the phagocytic functions of other receptors[68]. In line with these data, we found that N9 and primary microglia phagocytosis of Aβ is dramatically increased in the presence of anti-Aβ antibodies obtained from vaccinated dams. This effect is antibody-specific, as serum from sham-vaccinated dams (containing high titers of anti-HBSAg Abs) did not yield an equivalent increase. Importantly, antibody-mediated phagocytosis is a Syk-dependent process, as inhibition of Syk abrogated phagocytic increases by Abs in both N9 and primary cells. Taken together, the association of Aβ clearance with microglial Syk activation maternal vaccination in vivo, and the establishment of causal relations between MAbs and phagocytosis capacity in vitro serve as evidence for the modulatory effect of MAbs on microglial cellular function in eliminating Aβ from the brain parenchyma.

We thus propose that during a short and limited period, maternal vaccination had set the infrastructure for facilitating future phagocytosis and clearance of Aβ by microglia. Such infrastructure consists of upregulation of effector FcγR in the brain. This Abs-induced process is supported by a positive regulation loop of Abs-FcRn that facilitates Ab delivery via the placenta and lactation and increases Abs functionality. Upon Aβ-expression, this infrastructure facilitates Fc-mediated phagocytosis and Aβ clearance, without the addition of active vaccination. Of note, the 5xFAD mouse model of EOAD used here does not provide a good model for DS in terms of construct validity, due to several reasons: (a) the model is based on *APP* and *PSEN1* mutations rather than on *APP* triplication, (b) the 5xFAD model lacks triplication of non-*APP* Hsa21 genes, and (c) the mutations in *PSEN1*, which is not characteristic of DS, may result in a confounding effect. Furthermore, the 5xFAD lacks tau-related mutations, and resultantly, no tau hyperphosphorylation and NFT formation occur in this model. Nonetheless, its face validity in recapitulating early Aβ plaque pathology, as occurs in DS, is satisfactory at the phenotype level compared with other DS mouse models. To test the translatability of maternal vaccination to clinical trials, this approach needs to be tested in mouse models encompassing trisomy of Hsa21-orthologous genes, including human *APP*.

In this study, we tested the efficacy of maternal vaccination in reducing Aβ-related neuropathology and cognitive decline. We found that maternal vaccination reduced Aβ pathology dramatically independently of active vaccination, in a long-lasting manner, months after MAbs were undetectable. A combination of maternal and active vaccination yielded the most potent results in reducing cerebral Aβ levels and ameliorating cognitive decline. Mechanistically, we propose that maternal vaccination activates the FcR-Syk-cofilin axis, resulting in actin regulation and promotion of Aβ clearance by microglial cells. In vitro phagocytosis was indeed facilitated by MAbs, via Syk-dependent pathways. Thus, maternal vaccination may provide a novel therapeutic approach for preventing early Aβ accumulation and dementia, as occurs in DS individuals and some variants of fEOAD.

## Methods

**Study design**. 8-week old C57BL6 WT female mice (*n* = 10/group) were actively vaccinated using the AβCoreS DNA-vaccine (Fig. 1a). Females were then crossed with 5xFAD males. IgG transfer via the placenta and lactation were assessed in one cohort (*n* = 10/group), while long-term effects of maternal and active vaccination

were tested in a second cohort (*n* = 5/group). At 1 m, active vaccination against Aβ or sham was administered to maternally or sham-vaccinated offspring, to yield four experimental groups: sham-vaccinated M−/A−, maternally vaccinated M+/A−, actively vaccinated M−/A+, and combined maternal and actively vaccinated mice M+/A+. Sham-vaccinated WT mice were used as healthy controls.

**Animals**. The 5xFAD EOAD mouse model (Jackson Laboratories #34840), which encompasses five AD-related mutations within the *APP* and *PSEN1* genes[36], was used in this study to model early Aβ accumulation in DS. 5xFAD mice were generously provided by Michal Schwartz (The Weizmann Institute, Rehovot, Israel). C57BL mice were used as dams and healthy controls (Jackson Laboratories #000664, Bar Harbor, ME). Animal care and experimental procedures followed the NIH Guide for the Care and Use of Laboratory Animals and were approved by the Bar-Ilan University Animal Care and Use Committee.

**AβCoreS vaccine**. AβCoreS is based on the pVAX1 expression vector[27] and encodes the N-terminus-Aβ$_{1-11}$ fused to a Hepatitis-B surface antigen (HBsAg) and the Hepatitis-B core antigen (HBcAg), which acts to facilitate Ab production[69]. As a control treatment, an expression vector (pUC19, New England Biolabs) containing HBsAg was used.

**Vaccine administration**. Mice were intramuscularly injected at three time-points (14-day intervals), with 25 μg DNA (50 μl)[27]. Full electroporation configurations are detailed in the Supplementary Information section.

**Antibody titer**. Anti-Aβ$_{1-11}$ Ab serum levels were quantified using a standard indirect ELISA against recombinant Aβ$_{1-11}$ peptide. A detailed protocol is found in the Supplementary Information section.

**Exploratory behavior**. Exploratory behavior was recorded using a 40x40cm open field (OF) arena. The outer 8 cm were defined as the area periphery and the 24x24cm inner square as the center. Illumination was kept at 1300lux. Mice were allowed to explore the arena for 5 min freely[38]. Analysis of animal behavior in this task, as well as the following tasks, was conducted using Anymaze (Stoelting).

**Anxiety assessment**. Anxiety-related behavior was monitored using the elevated zero maze (EZM), a ring-shaped 65cm-high table, divided into closed and opened sections. The ring is 7 cm wide and has an outer diameter of 60 cm. The closed sections are confined by 20cm-high walls and a semi-transparent ceiling, whereas the opened sections have 0.5 cm high curbs at the edges. Illumination was kept at 1300lux, and the trial duration was 5min[39].

**Short-term object recognition memory**. Short-term memory was assessed using the novel object recognition (NOR) test[37,70]. Briefly, mice were placed in a 40x40cm arena with two different objects. In an acquisition trial, mice were allowed to explore their environment. In the following test trial, one of the objects was replaced by a novel object. Time spent near each of the objects was measured, and a preference index was calculated as the difference between time spent in the novel and familiar objects, divided by the total time spent near both objects.

**Spatial short-term memory**. We utilized a variant of the T-maze alternation test[71]. Full apparatus settings are found in the Supplementary Information section.

**Brain sample collection**. Mice were anesthetized using Ketamine-Xylazine (100 mg/kg, Vetoquinol, France, 10 mg/kg, Eurovet, The Netherlands, respectively) and perfused with PBS. For Histology, hemibrains were transferred to 4% paraformaldehyde (PFA) at 4 °C for 48 h. Following fixation, tissues were transferred to a gradient of 20% and 30% sucrose aqueous solutions for 24 h each. Hemibrains were then dissected into 40 μm-thick slices using a microtome and stored in a cryoprotectant solution (30% glycerol and 35% ethylene glycol) at -20 °C. For biochemical analysis, the cerebral cortex and hippocampi were separated, frozen on dry ice, and stored at −80 °C.

**Measuring Aβ$_{40/42}$ levels using sELISA**. Aβ$_{40}$ and Aβ$_{42}$ in the cortex were measured using a modification of a previously published sandwich-ELISA protocol[40]. The full protocol is found in the Supplementary Information section.

**Immunofluorescence**. 40μm-thick hemibrains were rinsed 5 times in 0.1% PBS-Triton for 5 min. Nonspecific bindings were blocked using 20% normal horse serum in PBS-T for 1 h at RT. For Aβ staining, antigen retrieval was conducted using incubation with 75% formic acid for 2 min at RT. For pSyk, antigen retrieval was conducted using sodium citrate buffer (pH = 6) for 20 min at 95 °C. Primary Abs for the following antigen were applied and incubated overnight at 4 °C: Aβ$_{42}$, Iba1, NeuN, GFAP, FcγRI, FcγRIIb, FcγRIII, FcγRIV, CD68, and pSyk. Next, sections were rinsed five times in PBS-T for 5 min, and fluorescence-tagged secondary Abs were applied for 1 h at RT. Primary and secondary Ab details are found

in Supplementary Table 1. Slices were then stained with Hoechst 33342 (H3570, Invitrogen, Carlsbad, CA) and diluted at 1:1,000, followed by five 5-min rinses with PBS-T.

**Aβ Plaque quantification**. Cerebral Aβ plaques were quantified computationally using blob detection tools in MATLAB (MathWorks, Natick, MA). Briefly, montage immunofluorescence images of the hippocampus and cortex were obtained using the X10 objective of a Leica DM6000 microscope (Leica Microsystems, Wetzlar, Germany), coupled to a controller module and a high sensitivity 3CCD video camera system (MBF Biosciences, VT) and an Intel Xeon workstation (Intel). Automated imaging was implemented using the Stereo Investigator software package (MBF Biosciences, VT). Analyzed brain sections spanned from −1.355 mm to −2.88 mm from Bregma. A total of six sections, every 7th to 8th section (280–320μm apart), were stained, imaged, and fed to a pre-validated plaque detection algorithm. Automated plaque quantification was optimized and validated prior to use by comparing the results of a validation set to the results from three independent human counting.

**Quantification of microglial markers**. Hippocampus and cortex were outlined according to the Paxinos atlas of the mouse brain. An average of 15 sections, spanning from 1.055 mm to −3.68 mm from Bregma, were used. Single-cell Iba1, FcγRI, CD68, and pSyk fluorescence intensity were filtered for noise and calculated using MATLAB (MathWorks, Natick, MA).

**RT-qPCR**. Cerebral Gene-expression was measured using standard TRIzol RNA extraction, cDNA generation, and RT-qPCR protocols. A detailed method is found in the Supplementary Information section. Primers used are detailed in Supplementary Table 1.

**Western blot**. A standard western blotting protocol was applied to measure hippocampal protein and phospho-protein. A detailed protocol is found in the Supplementary Information section. Primary and secondary Ab details are found in Supplementary Table 1.

**N9 microglial cell-line culture**. Murine embryonic microglia cell-line N9[53] were grown in Dulbecco's modified Eagle's medium (DMEM), supplemented with 10% fetal bovine serum (FBS), penicillin, streptomycin, and L-glutamine for 3-4 days to reach confluence.

**Murine primary microglia culture**. Adult primary microglia were harvested and cultured from 2-month-old C57BL6/j mice[72]. Briefly, mice were anesthetized using Ketamine-Xylazine (100 mg/kg, Vetoquinol, France, 10 mg/kg, Eurovet, The Netherlands, respectively) and perfused with cold Hanks' Balanced Salt Solution (HBSS, 14175, Thermo-Fisher Scientific, Waltham, MA). Brains were minced with a scalpel in an enzymatic solution containing papain and were incubated for 90 min at 37 °C, 5% CO$_2$. After 90 min, the enzymatic reaction was quenched with 20% FBS in HBSS, and cells were centrifuged for 7 min at 200$g$. The pellet was resuspended in 2 ml of 0.5 mg/ml DNase in HBSS and incubated for 5 min at room temperature. The tissue was gently disrupted, filtered through a 70mm-cell strainer, and centrifuged at 200$g$ for 7 min. The pellet was resuspended in 20 ml of 20% isotonic Percoll in HBSS, and pure HBSS was carefully overlaid on top of the cells-Percoll layer, followed by centrifuging at 200$g$ for 20 min with slow acceleration and no brakes. The pellet, containing the mixed glial cell population, was washed once in HBSS and suspended in Dulbecco's Modified Eagle's/F12 medium (DMEM/F12), supplemented with 10% FBS, penicillin, streptomycin, L-glutamine, and 10 ng/ml of carrier-free murine recombinant granulocyte and macrophage colony-stimulating factor (GM-CSF, 415-ML, R&D systems, Minneapolis, MN). Cell suspension from one brain was plated in a T75-cell culture flask coated with poly-D-lysine (P7280, Sigma, St. Louis, MO) and maintained at 37 °C, 5% CO$_2$ humidified incubator. The medium was changed twice a week. When full confluency was reached (after ~2 weeks), the floating cell layer, which includes pure microglia, was collected and centrifuged for 5 min at 400$g$. The pellet was then plated without GM-CSF on glass coverslips on 24-well plates for phagocytosis assay.

**Syk inhibition**. Highly selective Syk inhibitor BAY-61-3606 (B9685, Sigma, St. Louis, MO)[54], dissolved in 80% DMSO, was applied to N9 cells at different concentrations, ranging from 0.75 to 5 μM for 2 h, followed by the addition of aggregated human Aβ$_{42}$ peptide at a concentration of 750 nM (ab120301, Abcam, Cambridge, UK). Briefly, Aβ peptide was diluted in DMEM to a concentration of 7.5 mM and incubated for 1 h at 37 °C to induce aggregation. Inhibition of Syk and downstream signaling molecules were assessed using western blotting.

**Protein extraction from cell-line and primary microglia**. Cells were lysed in 0.1% SDS RIPA buffer (150 mM NaCl, 5 mM EDTA, 50 mM Tris-base, 1% Triton, 0.5% Na-deoxycholate, 0.1% SDS in aqueous solution) containing protease inhibitor (1:100, P2714, Sigma, St. Louis, MO) and phosphate inhibitor (1:100, P5726,

Sigma, St. Louis, MO) cocktails and incubated for 30 min on ice, then centrifuged at 17,000 g for 20 min. The supernatant was separated, and total protein concentration was measured using the BCA method.

**Fluorescent beads phagocytosis assay**. N9 cells were deprived of FBS and incubated this Syk inhibitor BAY-61-3606 (5 μM) for 3 h. 1μm-fluorescent beads (L1030, Sigma, St. Louis, MO) were pre-coated with FBS for 1 h at 37 °C, then applied to cells at a dilution of 1:1000 for 1 h at 37 °C. Cells were then washed five times with PBS (containing calcium and magnesium) and trypsinized. Phagocytic FITC + cell count was conducted using the BD-LSRFortessa cell analyzer (BD bioscience, East Rutherford, NJ).

**Aβ phagocytosis assay**. Aβ$_{42}$ peptide (ab120301, Abcam, Cambridge, UK) was diluted in DMEM and incubated for 3 h at 37 °C to induce aggregation. Aggregated Aβ was incubated with serum from anti-Aβ vaccinated mice, sham-vaccinated mice, or DMEM (500 ng/ml antibody, determined by ELISA) for 1 h at 37 °C with gentle agitation, to produce Ab–Ag complex. N9 and primary cells were treated with 5 μM BAY-61-3606 for 3 h for Syk inhibition and incubated with aggregated Aβ with or without the addition of serum for 4 h. Next, cells were washed five times with PBS (containing calcium and magnesium). For immunofluorescence, cells were fixed with 4% PFA and permeabilized using 0.3% Triton in PBS. Next, cells were stained with anti-CD68 and anti-Aβ1-14 primary antibodies, and probed with appropriate Alexa-fluor secondary antibodies (Supplementary Table. 1). Unbiased stereological counting of Aβ foci was performed using the Stereo-investigator software with a counting frame of 300×300 μm. For FACS analysis, cells were fixed with 4% PFA, permeabilized using 0.3% Triton in PBS, and stained for intracellular Aβ using an anti-Aβ1-14 antibody (Supplementary Table. 1). Aβ-containing cell count was conducted using the BD-LSRFortessa cell analyzer (BD bioscience, East Rutherford, NJ).

**Statistics and reproducibility**. The data presented as mean ± SEM were tested for significance in the unpaired $t$-test for samples of equal variance, one-way ANOVA, repeated measures (RM) two-way ANOVA, two-sample Kolmogorov–Smirnov test, Pearson's correlation coefficient, or the $\chi^2$ test for independence. Post-hoc tests were conducted using the Tukey or Bonferroni corrections. All error bars represent SEM were calculated as $\frac{std(x)}{\sqrt{n}}$ for numeric variables, and as $\sqrt{\frac{p(1-p)}{n}}$ for binomial variables. Outliers were identified using the robust regression and outlier removal (ROUT) method with coefficient $Q = 1\%$[73]. Sample size: (a) vaccination of dams ($n = 10$ per group), (b) transplacental and breastfeeding IgG transfer ($n = 10$ per group, $n = 5$ for control WT), (c) long-term effect of maternal and active vaccination ($n = 5$ per group). qPCR, ELISA, IF, and FACS experiments were repeated in triplicates, WB experiments were conducted with five biological replicates. Naïve dams were randomly allocated to control/vaccine groups. Maternally/control vaccinated offspring were not randomly allocated to control/active vaccination to maintain original littermates in the same cages. All behavioral tests were conducted under experimenter blinding to group identity. Significant results were marked according to conventional critical P-values: $*P < 0.05$, $**P < 0.01$, $***P < 0.001$, $****P < 0.0001$.

**Reporting summary**. Further information on research design is available in the Nature Research Reporting Summary linked to this article.

## Data availability
Source data behind the graphs are available on the Open Science Framework: https://osf.io/jnxrz/. All the data supporting the findings of this study are freely available upon request.

## Code availability
All the MATLAB codes are used in this study are freely available upon request.

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

## Acknowledgements

This study was conducted in the Paul Feder laboratory of Alzheimer's disease research and was supported by the Clore Israel Foundation and in part with the Intramural Research Program of the National Institute on Aging. Yael Laure edited the manuscript.

## Author contributions

T.I. and E.O. conceptualized and designed this study. T.I., R.N., and L.B.S. performed the experiments. T.I. analyzed the data. R.M. provided methodological and biological insights and technical aid. A.B. developed the AβCoreS vaccine and provided mechanistic insights. T.I. and E.O. wrote the manuscript. All authors reviewed and approved the final version.

## Competing interests

The authors declare no competing interests.
