## [Peer Review File · Communications Biology]

Reviewers' comments:

Reviewer #1 (Remarks to the Author):

This is a very well-written, designed and executed manuscript. The authors propose to investigate if maternal immunization could reduce amyloid-beta deposits and consequent cognitive decline. To test their hypothesis, they vaccinated WT female mice with a DNA vaccine expressing AB1-11, followed by mating with 5xFAD males. The results demonstrated a reduction in cortical AB levels, improvements in short-term memory, and the participation of a mechanism involving activation of FcγR1/Syk/Cofilin pathways. Statistical analyses and controls are appropriate and the methods are described in a way that allows reproducibility.

My main concern is with the stretching of the animal model of choice as a model for DS. The introduction is heavily based on using 5x animals as a model for DS. 5x mice recapitulates the early deposition of Abeta and that is also seen in DS; however, the absence of PSEN mutation in DS could potentially be a confounder on the current findings. Without directly comparing these results with other DS animal models it is not possible to say that maternal vaccination might reduce the Abeta pathology in DS, as the authors ultimately are hoping to say. The text should be focused on the mechanisms found and their importance for early AB deposition, which might have implications in EOAD, DS, etc.

Minor concern:

The Iba1 staining on Figures 4 and 5 doesn't look like a typical staining for microglia cells. DAPI staining is also barely visible on Figure 5. This might be a poor choice of representative images, and it should be fixed to guarantee that the results being observed are what they seem like.

Reviewer #2 (Remarks to the Author):

The present work aimed to evaluate maternally-transferred anti-Aβ antibodies' potential over cognitive parameters associated with Alzheimer's disease (AD) symptoms in Down syndrome (DS) individuals by using an early-onset AD (EOAD) transgenic mice as a model (5xFAD). The vaccine employed is a DNA-vaccine expressing Aβ1-11, which showed a lack of toxic T response stimulation. The studied groups include maternal-only, actively postnatal vaccinated-only, and maternal and actively postnatal vaccinated animals, besides the WT and/or sham animal groups. The overall work conclusion proposes an effect of maternal antibodies in reducing Aβ species in vaccinated mice through activation of microglial phagocytosis via the FCγR1/Syk/Cofilin pathway. Cognitive parameters were evaluated at four months of age, suggested to improve in some tests.

1. The work shows an interesting approach to preventing Aβ pathology early in life during embryonic development. However, DS individuals present the triplication of chromosome 21, where it is localized the APP gene, and other 600 genes (10.1038/s41582-018-0132-6). In this regard, other genes that are not overexpressed in the 5xFAD could be contributing to the dementia pathology in DS people (10.1093/brain/awy159). Also, AD-associated pathology in DS individuals includes, in addition to the Aβ accumulation, NFTs formation, which is not represented in the mouse model used in the study. The authors should discuss these aspects regarding their results and projections.

2. Regarding the abstract and overall conclusions, the work claims, "maternal immunization can alleviate AB-mediated decline in individuals with DS." It is suggested that authors change "DS individuals" to "experimental animals" or something similar - since the study uses mice. The overall cognitive parameters results show a small if no differences between maternal versus actively

vaccinated experimental mice. According to the results, early active-vaccination appears enough to improve cognition, despite a positive effect on A β clearance and microglial phagocytosis. The authors should disclose these differences in the results section and complement the discussion in this aspect. In Figure 3I, which is the red staining? Legend has no information about it.

3. About the hippocampi slices in Figure 4K, they are from the same region? Some of the hippocampi slices seem to be from the anterior and other from the posterior hippocampi, and microglia distribution – and A β plaque burden – can be different from regions.

4. Interesting results were found in the signal transduction results. However, differences in total and phosphorylated proteins' expression should not be interpreted individually. The difference reflecting function is the ratio of phospho/total protein. In this regard, pSyk/Syk or pCofilin/Cofilin ratios show no differences between groups (Figure 5B and 5E, respectively). Considering these changes, how can authors interpret the following results – pAKT/AKT and pERK/ERK - on the phagocytosis activity observed?

5. The results presented with serum incubation from vaccinated individuals are notable. However, why did authors not use serum from offspring instead of dams mothers, given the work's novelty regarding the maternal-vaccination effect?

6. Given the recent importance of TREM2 genetic variance on AD (10.1126/science.abb8575), their role in A β clearance, what authors would expect from TREM2 expression and function after vaccination types?

7. Inline 21, from page 22 (Discussion section), authors refer to the tolerability of DNA-vaccination in "patients" using mice vaccination reference. Please, change the writing or include a reference from human patients.

8. Supplementary Figure 1 shows a full image of the immunological characterization of both vaccination procedures. A recommendation is to include it as a primary figure. Supplementary Figure 9 could also be included as the last primary figure, summarizing the findings and the main differences between vaccination times/types.

Minor comments:

1. The last paragraph of the introduction refers to figures as a part of the Results sections. Although a summary of results is required, no reference to figures should be included in the introduction section.

2. The Results section is complete, with details for the p values and average \pm , etc. However, these details difficult the reading flow. A suggestion is to incorporate these details in the legends, which in some of them are incomplete, reducing to the text the significance and p values.

3. The font used in the figures are not standardized, and some of them are too small, making difficult the understanding – such as Figure 1C, D; L-Q; Figure 3G, Figure 4L and N; Figure 6K on the right.

4. The format of the Western blot bands is different among figures. The suggestion is to follow the Figure 6A format, which is clear and organized.

Rebuttal letter

*Responses to the reviewers' comments are highlight in blue.

**Changes in the manuscript are highlighted in blue in the revised manuscript file, and are included in this letter as well.

Reviewer #1 (Remarks to the Author):

This is a very well-written, designed and executed manuscript. The authors propose to investigate if maternal immunization could reduce amyloid-beta deposits and consequent cognitive decline. To test their hypothesis, they vaccinated WT female mice with a DNA vaccine expressing AB1-11, followed by mating with 5xFAD males. The results demonstrated a reduction in cortical AB levels, improvements in short-term memory, and the participation of a mechanism involving activation of FcγR1/Syk/Cofilin pathways. Statistical analyses and controls are appropriate and the methods are described in a way that allows reproducibility.

1. My main concern is with the stretching of the animal model of choice as a model for DS. The introduction is heavily based on using 5x animals as a model for DS. 5x mice recapitulates the early deposition of Abeta and that is also seen in DS; however, the absence of PSEN mutation in DS could potentially be a confounder on the current findings. Without directly comparing these results with other DS animal models it is not possible to say that maternal vaccination might reduce the Abeta pathology in DS, as the authors ultimately are hoping to say. The text should be focused on the mechanisms found and their importance for early AB deposition, which might have implications in EOAD, DS, etc.

Response:

We thank the reviewer for the positive critique of our work. The available DS mouse models are quite complex with respect to interpretation of inter-gene effects, as each model involves the trisomy of a different number of genes in addition to Amyloid Precursor Protein (APP). For example, Ts65Dn, the most widely used murine model of DS, exhibits a translocation that results in an extra small chromosome that holds the Mmu16 region containing 92 genes orthologous to Hsa21 (*App-Zbtb21*). These mice are trisomic for about two-thirds of the genes orthologous to Hsa21, but this additional chromosome also carries genes originating from the Mmu17 that are not related to DS, including 46 protein-coding genes. In another example, Ts1Cje mice do not carry an extra chromosome but are trisomic 81 genes orthologous to Hsa21 (*Sod1-Mx1*). However, a translocation between the Mmu16 proximal to *Sod1* and the very distal region of the Mmu12 occurred, with the Mmu16 breakpoint being between *App* and *Sod1*. The humanized Tc1 mouse model contains an additional Hsa21 chromosome and is functionally trisomic for ~120 protein-coding genes, and most of the human Hsa21 genes are likely expressed at the mRNA, protein, and functional levels. Unfortunately, it was revealed that multiple structural rearrangements and mutations in genes of interest (such as *APP*) occurred in these mice, likely caused by gamma irradiation during the generation of the model. As a result, the *APP* gene in Tc1 mice is not functional. In addition, Tc1 mice present variable levels of mosaicism of the extra chromosome in different tissues, as the extra chromosomes seem to be randomly lost, confounding the analysis of phenotypic consequences^{1,2}.

Furthermore, since current models exhibit mouse *APP* trisomy, in which the mouse APP protein poorly aggregates compared with human APP, we sought to utilize a mouse model that recapitulates the early expression of high levels of human APP, which forms plaques, as occurs in DS. However, we agree that the 5xFAD mouse model does not recapitulate the entire milieu of DS pathology but rather APP-related pathology that results in early onset of amyloid

pathology. In this manner, 5xFAD mice provide a higher face validity than any other existing mouse model of DS.

Therefore, in accordance with this comment, we have toned down the implications of the described findings such that they might be applicable to pathologies that exhibit early A β deposition, such as EOAD and DS.

Among the changes made in the revised manuscript, we would like to emphasize the following changes:

- Abstract, page 2, line 4: “[...] such as Alzheimer’s disease (AD) and AD-pathology in Down Syndrome (DS).”
- Abstract, page 2, line 5: “While familial early-onset AD (fEOAD) is associated with autosomal dominant mutations in the *APP*, *PSEN1,2* genes, that promotes cerebral Amyloid- β (A β) deposition [...]”.
- Abstract, page 2, line 17: “These data suggest that maternal immunization can alleviate cognitive decline mediated by early A β deposition, **as occurs in EOAD and DS.**”
- Introduction, page 3, line 2 – page 4, line 2: The introduction opens with an exposition on maternal immunity rather than on DS.
- Introduction, page 3, line 18: EOAD, and the L166P mutation of PSEN1, in particular, were added as candidate conditions that can be alleviated by maternal vaccination.
- Introduction, page 5, line 18: “[...] occurs in fEOAD and DS.”
- Introduction, page 6, line 13: “Indeed, the 5xFAD model has low construct validity in modeling DS, due to several reasons: (a) the model is based on *APP* and *PSEN* mutations rather than on *APP* triplication, (b) the 5xFAD model lacks triplication of non-APP Hsa21 genes, and (c) **the mutations in *PSEN1*, which is not characteristic**

of DS, may result in a confounding effect. Furthermore, the 5xFAD lacks tau-related mutations and resultantly, no tau hyperphosphorylation and NFT formation occur in this model.”

- A complementing paragraph on the shortcomings of using the 5xFAD strain was also added to the Discussion section (page 26, line 24), to disclose this topic fully:

“Of note, the 5xFAD mouse model of EOAD used here does not provide a good model for DS in terms of genetics. Since the model is based on APP mutations rather than on APP triplication, it lacks the triplication of non-APP Hsa-21 genes and tau-related mutations and harbors potential confounding effects related to the PSEN1 mutations. Nonetheless, early A13 plaque pathology in 5xFAD mice better recapitulates DS A13 pathology compared with other DS mouse models.”

- A recommendation to test this strategy in a classical model of DS was added to page 27, line 6: “To test the translatability of maternal vaccination to clinical trials, this approach needs to be tested in mouse models encompassing trisomy of Hsa21-orthologous genes, including human APP.”

Minor concern:

2. The Iba1 staining on Figures 4 and 5 doesn't look like a typical staining for microglia cells. DAPI staining is also barely visible on Figure 5. This might be a poor choice of representative images, and it should be fixed to guarantee that the results being observed are what they seem like.

Response:

We thank the reviewer for this comment. We have now generated new high-magnification images for Figures 4K and 5H. Additionally, in accordance with the reviewer's comment,

DAPI exposure was enhanced for better clarity. The microglial staining represents the morphology of plaque-adjacent microglial cells, although all microglial cells of the hippocampus were included in the quantitative analysis. As was previously shown, microglia associated with A β plaques exhibit altered transcriptomic signature and proteomic phenotype compared with microglia that are more distal from plaques and microglia from WT mice³⁻⁵. This cell type, also termed Disease-associated microglia (DAMS) or microglia in neurodegeneration (MGnD), has an atypical morphology as cells are more amoeboid and tend to be grouped in clusters around plaques⁶, compared with ramified homeostatic microglia that are distal from plaques. To illustrate this difference, in accordance with the reviewer's comment, we have added Supplementary Fig. S8, showing homeostatic ramified microglia from the hippocampi of naïve, maternally, and actively vaccinated mice (see figure below).

Fig. S8. CD68 expression following maternal and active vaccination in ramified homeostatic microglia. (A) Double labeling of CD68 and Iba1⁺ in ramified microglia, taken using the x63 objective.

The following text was added to the Results section:

Page 18, line 14: “High magnification exemplars presenting hippocampal CD68-expressing microglia (Fig. 5H). Ramified microglia were also observed in the hippocampi of naïve, maternally, and actively vaccinated mice (Supplementary Fig. S8A).”

Reviewer #2 (Remarks to the Author):

The present work aimed to evaluate maternally-transferred anti-A13 antibodies' potential over cognitive parameters associated with Alzheimer's disease (AD) symptoms in Down syndrome (DS) individuals by using an early-onset AD (EOAD) transgenic mice as a model (5xFAD). The vaccine employed is a DNA-vaccine expressing A131-11, which showed a lack of toxic T response stimulation. The studied groups include maternal-only, actively postnatal vaccinated-only, and maternal and actively postnatal vaccinated animals, besides the WT and/or sham animal groups. The overall work conclusion proposes an effect of maternal antibodies in reducing A13 species in vaccinated mice through activation of microglial phagocytosis via the FC γ R1/SYK/COLLIN pathway. Cognitive parameters were evaluated at four months of age, suggested to improve in some tests.

3. The work shows an interesting approach to preventing A13 pathology early in life during embryonic development. However, DS individuals present the triplication of chromosome

21, where it is localized the *APP* gene, and other 600 genes (10.1038/s41582-018-0132-6). In this regard, other genes that are not overexpressed in the 5xFAD could be contributing to the dementia pathology in DS people (10.1093/brain/awy159). Also, AD-associated pathology in DS individuals includes, in addition to the A β accumulation, NFTs formation, which is not represented in the mouse model used in the study. The authors should discuss these aspects regarding their results and projections.

Response:

We thank the reviewer for these important insights. The available DS mouse models are quite complex with respect to interpretation of inter-gene effects, as each model involves the trisomy of a different number of genes in addition to Amyloid Precursor Protein (APP). For example, Ts65Dn, the most widely used murine model of DS, exhibits a translocation that results in an extra small chromosome that holds the Mmu16 region containing 92 genes orthologous to Hsa21 (*App-Zbtb21*). These mice are trisomic for about two-thirds of the genes orthologous to Hsa21, but this additional chromosome also carries genes originating from the Mmu17 that are not related to DS, including 46 protein-coding genes. In another example, Ts1Cje mice do not carry an extra chromosome but are trisomic 81 genes orthologous to Hsa21 (*Sod1-Mx1*). However, a translocation between the Mmu16 proximal to *Sod1* and the very distal region of the Mmu12 occurred, with the Mmu16 breakpoint being between *App* and *Sod1*. The humanized Tc1 mouse model contains an additional Hsa21 chromosome and is functionally trisomic for ~120 protein-coding genes and most of the human Hsa21 genes are likely expressed at the mRNA, protein and functional levels. Unfortunately, it was revealed that multiple structural rearrangements and mutations in genes of interest (such as *APP*) occurred in these mice, likely caused by gamma irradiation during the generation of the model. As a result, the *APP* gene in Tc1 mice is not functional. In addition, Tc1 mice present variable levels of mosaicism of the extra chromosome in different tissues, as the extra chromosomes seem to be randomly lost, confounding the analysis of phenotypic consequences^{1,2}.

Furthermore, since the current models exhibit mouse *APP* trisomy, which fails to aggregate, we sought to utilize a mouse model that recapitulates the early expression of high levels of APP, which forms plaques, as occurs in DS. However, we agree that the 5xFAD mouse model does not recapitulate the entire milieu of DS pathology but rather APP-related pathology that results in early onset of amyloid pathology. In this manner, 5xFAD mice provide a higher face validity than any other existing mouse model of DS.

Therefore, in accordance with this comment, we have toned down the implications of the described findings such that they might be applicable to pathologies that exhibit early A β deposition, such as EOAD and DS.

Among the changes made in the revised manuscript (marked in blue in the manuscript file), we would like to emphasize the following changes:

- Abstract, page 2, line 4: “[...] such as Alzheimer’s disease (AD) and AD-pathology in Down Syndrome (DS).”
- Abstract, page 2, line 5: “While familial early-onset AD (fEOAD) is associated with autosomal dominant mutations in the *APP*, *PSEN1,2* genes, that promotes cerebral Amyloid- β (A β) deposition [...]”.
- Abstract, page 2, line 17: “These data suggest that maternal immunization can alleviate cognitive decline mediated by early A β deposition, **as occurs in EOAD and DS.**”
- Introduction, page 3, line 2 – page 4, line 2: The introduction opens with an exposition on maternal immunity rather than on DS.
- Introduction, page 3, line 18: EOAD, and the L166P mutation of PSEN1, in particular, were added as candidate conditions that can be alleviated by maternal vaccination.
- Introduction, page 5, line 18: “[...] occurs in fEOAD and DS.”

- Introduction, page 6, line 13: “Indeed, the 5xFAD model has low construct validity in modeling DS, due to several reasons: (a) the model is based on *APP* and *PSEN1* mutations rather than on *APP* triplication, (b) the 5xFAD model lacks triplication of non-*APP* Hsa21 genes, and (c) the mutations in *PSEN1*, which is not characteristic of DS, may result in a confounding effect. **Furthermore, the 5xFAD lacks tau-related mutations and resultantly, no tau hyperphosphorylation and NFT formation occurs in this model.**”
- A complementing paragraph on the shortcomings of using the 5xFAD strain was also added to the Discussion section (page 26, line 24), to disclose this topic fully: “Of note, the 5xFAD mouse model of EOAD used here does not provide a good model for DS in terms of genetics. Since the model is based on *APP* mutations rather than on *APP* triplication, it lacks the triplication of non-*APP* Has-21 genes and tau-related mutations and harbors potential confounding effects related to the *PSEN1* mutations. Nonetheless, early A13 plaque pathology in 5xFAD mice better recapitulates DS A13 pathology compared with other DS mouse models.”
- A recommendation to test this strategy in a classical model of DS was added to page 27, line 6: “To test the translatability of maternal vaccination to clinical trials, this approach needs to be tested in mouse models encompassing trisomy of Hsa21-orthologous genes, including human *APP*. ”

4. Regarding the abstract and overall conclusions, the work claims, "maternal immunization can alleviate AB-mediated decline in individuals with DS." It is suggested that authors change "DS individuals" to "experimental animals" or something similar - since the study uses mice.

Response:

We agree with the reviewer's comment and have changed this sentence accordingly:

- Abstract, page 2, line 10: “We hypothesized that maternally transferred anti-A13 antibodies might promote the removal of early A13 accumulation in the central nervous system **of human APP-expressing mice.**”
 - Abstract, page 2, line 12: “To this end, a DNA-vaccine expressing A13¹⁻¹¹ was delivered to wild-type female mice, followed by mating with 5xFAD males, **which exhibit early A β plaque formation.**” (The previous “similar to individuals with DS” was deleted).
 - Abstract, page 2, line 17: “These data suggest that maternal immunization can alleviate cognitive decline mediated by early A13 deposition, as occurs in EOAD and DS.”
 - Conclusions, page 27, line 3: “Thus, maternal vaccination may provide a novel therapeutic approach for preventing early A13 accumulation and dementia **as occurs in DS individuals and some variants of fEOAD.**”
5. The overall cognitive parameters results show a small if no differences between maternal versus actively vaccinated experimental mice. According to the results, early active-vaccination appears enough to improve cognition, despite a positive effect on A13 clearance and microglial phagocytosis. The authors should disclosure these differences in the results section and complement the discussion in this aspect.

Response:

In accordance with the reviewer's comment, we have further emphasized the functional differences in outcome between maternal-only vs. active immunization and the combined intervention.

- Results, page 9, line 23: [...] “suggesting that maternal vaccination alone **only partially ameliorated deficits in short-term memory, while active vaccination and combined vaccination resulted in a full short-term memory restoration.**”

- Discussion, page 23, line 24 “Intriguingly, maternal vaccination alone had no effect on exploratory behavior and only a partial positive effect on short-term memory. In contrast, active vaccination yielded a full rescue of short-term memory ability.”
 - Conclusions, page 27, line 14: “A combination of maternal and active vaccination yielded the most potent results **in reducing cerebral A β levels and ameliorating cognitive decline.**”
6. In Figure 3I, which is the red staining? Legend has no information about it.

Response:

The color scheme in Figure 3I is aimed to separate between adjacent plaques. Colors were computationally added to Alexa-488-positive A13 plaques. It is now better clarified in the figure legend for Figure 3:

“(I) Plaque load was quantified using immunofluorescence and blob detection. **Plaques were pseudocolored in high magnification images to better separate and emphasize different plaques.**”

7. the hippocampi slices in Figure 4K, they are from the same region? Some of the hippocampi slices seem to be from the anterior and other from the posterior hippocampi, and microglia distribution – and A13 plaque burden – can be different from regions.

Response:

We thank the reviewer for this observation. New exemplar images were taken from comparable brain regions in Figure 4K. According to a previously published division of the hippocampus⁷, coronal brain slices were taken from the dorsal hippocampus, at position 285, coronal level 72 in the Allen Brain Atlas (-1.755 mm from Bregma, see reference image below). Additionally, DAPI exposure was enhanced for better clarity of the dentate-gyrus structure.

8. Interesting results were found in the signal transduction results. However, differences in total and phosphorylated proteins' expression should not be interpreted individually. The difference reflecting function is the ratio of phospho/total protein. In this regard, pSyk/Syk or pCofilin/Cofilin ratios show no differences between groups (Figure 5B and 5E, respectively). Considering these changes, how can authors interpret the following results – pAKT/AKT and pERK/ERK - on the phagocytosis activity observed?

We thank the reviewer for this comment. Indeed, when considering intracellular signaling pathways, the ratio between phospho- and total protein of interest is typically assessed, to reflect pathway activation/inactivation when the ratio is higher than the baseline. However, this approach is effective when the total unphosphorylated protein levels are constant between experimental groups, given that normalization to a constant protein was carried out. For example, our data suggest that total ERK and total AKT levels do not differ between maternally, actively, and naïve mice, whereas pERK and pAKT levels are elevated following active vaccination (Fig. 5A, C-D). These findings suggest that the FcγR-mediated phagocytosis pathway is mediated by phosphorylation of these signaling molecules rather than by upregulation of their expression. These data are indeed in compliance with previous reports^{8,9}.

However, when basic levels of an unphosphorylated signaling molecule differ between groups, as observed for Syk, it is essential to dissect the results at both total and phospho-protein levels. This is due to the possibility that the pathway of interest is regulated not only by phosphorylation but also via translational regulation of protein expression.

With respect to Syk, previous works suggest that Syk function is regulated by both protein expression and phosphorylation. In a mouse model of ischemia, both Syk and pSyk brain levels are elevated compared with controls. Additionally, Both Syk and pSyk are reduced after the administration of a Syk inhibitor¹⁰. More specifically, following oxygen/glucose deprivation of microglial cells, Syk and pSyk levels were both higher than in naive cells. Moreover, unphosphorylated Syk levels were reduced after inhibition of TREM-1 in brains of ischemic mice and in following oxygen/glucose-deprived microglia¹⁰. Similarly, a recent paper also found that total Syk is elevated following cerebral focal ischemia in mice¹¹. Lastly, in a mouse model of tauopathy, total Syk was found to be elevated compared with controls¹².

Taken together, our data, as well as previous works, imply that Syk is being regulated not only at the phosphorylation level but also at the protein expression level. Since Syk levels were normalized to β -Tubulin, we find that both Syk and pSyk are elevated following maternal vaccination, reflecting both the upregulation in Syk expression and subsequently the elevated levels of cellular pSyk. Fc γ RI was found to be upregulated in the brains and microglia of maternally vaccinated mice both at the transcript and expression levels (Fig. 4). Upregulation of downstream molecules such as Syk may suggest a common regulation of the receptor and its associated recruited molecules. Since both Syk and pSyk are elevated in maternally vaccinated mice, their ratio does not differ from those of non-maternally vaccinated mice. Nonetheless, for the reasons mentioned above, we find the separate analyses of Syk and pSyk to be meaningful.

Other cases that should be considered separately are signaling molecule whose unphosphorylated form does not reflect an absence of signal (i.e. ERK and AKT) but rather has its own cellular role,

which can be opposite to the phosphorylated form, as in the case of Cofilin. The actin-depolymerizing factor (ADF)/cofilin protein family consists of small actin-binding proteins that play central roles in accelerating actin turnover by disassembling actin filaments. The activity of cofilin is regulated by phosphorylation on residue Ser3 by LIM kinases (LIMK1 and LIMK2) and TES kinases (TESK1 and TESK2), which inhibit its interaction with actin. Cofilin binding to actin filaments promotes actin depolymerization, while the inactive pCofilin promotes actin filaments stability and elongation^{13,14}. Thus, both Cofilin and pCofilin are involved in actin dynamics, not just pCofilin. Moreover, the phosphorylation state of cofilin at Y68 promotes its ubiquitination and proteasomal degradation, suggesting that cofilin activity is regulated both by phosphorylation and phosphorylation-dependent ubiquitination–proteasome pathway¹⁵. For these reasons, we find the separate analyses of Cofilin and pCofilin to be meaningful.

The bars in Fig. 5B-F were re-ordered to maintain the western blot loading order, as presented in the blots.

9. The results presented with serum incubation from vaccinated individuals are notable.

However, why did authors not use serum from offspring instead of dams mothers, given the work's novelty regarding the maternal-vaccination effect?

Response:

Based on the peak in pups' antibody titer at P14 (IgG concentration of 30 µg/ml) and the fraction of antibodies reported to cross the BBB (0.1%)^{16,17}, the IgG concentration in their cerebral extracellular space is 30 ng/ml. To mimic these antibody levels in microglia culture plates, we added the appropriate amount of sera containing IgG. To get such a concentration in the volume of a cell-culture plate, we had to use the dams' serum, which initially had higher antibody

concentration than that of maternally vaccinated pups. It is technically impossible to extract the required blood amounts from newborns in order to obtain the required antibody conc/dose.

10. Given the recent importance of TREM2 genetic variance on AD (10.1126/science.abb8575), their role in A β clearance, what authors would expect from TREM2 expression and function after vaccination types?

We agree with the reviewer regarding the importance of TREM2 to microglial activation and A β clearance. Indeed, the TREM2 and Fc γ R-mediated phagocytosis pathways greatly overlap, as both receptors are associated with an ITAM domain, which upon activation recruits and binds Syk that subsequently undergoes autophosphorylation. Additionally, both pathways facilitate actin rearrangement crucial for phagosome formation¹⁸⁻²⁰. In accordance, we would expect the TREM2 pathway to be activated following Ab-induced microglial phagocytosis, as shown in the additional data below. DNAX-activating protein of 12 kDa (DAP12; also known as TYROBP and KARAP) is a signaling adapter protein expressed by various innate immune cells, including microglia. DAP12, which has an ITAM domain, associates with TREM2 and is essential for its function in signal transduction following ligand binding^{21,22}. Indeed, we found that the cerebral expression of *TYROBP*, the gene encoding DAP12, is upregulated in 1m-old maternally vaccinated mice compared with controls (supplementary Fig. S4G). Interestingly, elevation in *TYROBP* expression remained high by 5m of age in maternally vaccinated mice, whereas active vaccination and the combined treatment yielded a non-significant elevation (Supplementary Fig. S4H). We speculate that TREM2-mediated microglial activation is a secondary and complementary event, initiated following Fc γ R-upregulation and activation of its downstream signaling in microglia. This is since the main receptors for IgG antibodies are the FcRs family, not TREM2. Moreover, TREM2 was found to be required for antibody-A β complexes-induced phagocytosis (a process mediated by FcR) in the N9 mouse microglial cell line and primary microglia, showing TREM2 enables the phagocytic functions of other receptors²³. For these reasons, we recognize that TREM2 activation

may explain in part the outcomes observed after maternal vaccination. However because of the overlap in signaling pathways and because the proposed therapy is mediated by IgG2b, we chose to focus on the FcγR-phagocytosis pathway.

now as Supplementary Fig. S4 G-H, and the paragraphs below were added to the manuscript:

Results section, page 15, line 5:

“Triggering receptor expressed on myeloid cells-2 (TREM2), an innate immune receptor expressed on microglia is heavily implicated in microglial recruitment to Aβ plaques^{20,24,25}. DNAX-activating protein of 12 kDa (DAP12; also known as TYROBP) associates with TREM2 and is essential for its function in signal transduction following ligand binding^{21,22}. Intriguingly, 1m-old maternally vaccinated WT and 5xFAD mice exhibit high expression of *TYROBP* compared with controls ($P<0.01$, Supplementary Fig. S4G). *TYROBP* expression remained elevated months after maternal vaccination, as its expression in M+/A- mice was higher compared with M-/A- mice at 5m of age ($P<0.05$, Supplementary Fig. S4H). M-/A+ and M+/A+ exhibited a non-significant elevation in *TYROBP* expression compared with controls (Supplementary Fig. S4H). This suggests that in addition to upregulating FcγR, maternal vaccination also enhances the expression of *TYROBP* in a long-lasting manner, which supports TREM2 signaling in microglia.”

Discussion section, page 25, line 22:

“In addition to FcγR-Syk axis activation in microglia, we found that maternal vaccination upregulates the expression of TYROBP (DAPI2), a signaling adapter protein essential for TREM2 signaling in microglia^{21,22}. We speculate that TREM2-mediated microglial activation is a secondary and complementary event, initiated following FcγR-upregulation, as FcRs, rather than TREM2, are the primary receptor family for IgG antibodies. Moreover, TREM2 is required for the full phagocytic capacity of antibody-Aβ complexes (a process mediated by FcR for phagocytosis) by the N9 mouse microglial cell line and primary microglia, showing that TREM2 enables the phagocytic functions of other receptors²³.”

Figure legend:

(G) Cerebral levels of TYROBP following maternal vaccination at 1m of age (H) Cerebral levels of TYROBP following maternal and active vaccination at 5m of age. *P<0.05, **P<0.01, Pearson’s correlation, Two-way ANOVA.

Primers for TYROBP qPCR were added to Supplementary Table 1:

Gene	Forward primer	Reverse primer	Annealing temp (°C)	Product size (bp)
TYROBP	GGTGTGACTCT GCTGATTGC	AAGCTCCTGAT AAGGCGACTC	56	127

11. In line 21, from page 22 (Discussion section), authors refer to the tolerability of DNA-vaccination in "patients" using mice vaccination reference. Please, change the writing or include a reference from human patients.

The following reference, reporting unremarkable safety profile in clinical trials, was added on page 22, line 22²⁶:

[...] “reporting enhancement of DNA immunogenicity in humans and tolerability by **experimental animal and human subjects** without causing negative side effects^{26,27}.”

12. Supplementary Figure 1 shows a full image of the immunological characterization of both vaccination procedures. A recommendation is to include it as a primary figure. Supplementary Figure 9 could also be included as the last primary figure, summarizing the findings and the main differences between vaccination times/types.

In accordance with the reviewer’s suggestion, Supplementary Figure 1 was added to primary Figure 1 as panel R, and supplementary Figure 9 was added to primary Figure 5 as panel G.

Minor comments:

13. The last paragraph of the introduction refers to figures as a part of the Results sections. Although a summary of results is required, no reference to figures should be included in the introduction section.

In accordance with the reviewer’s comment, the reference to Figure 1A was removed from the last paragraph of the introduction section.

14. The Results section is complete, with details for the p values and average \pm , etc. However, these details difficult the reading flow. A suggestion is to incorporate these details in the legends, which in some of them are incomplete, reducing to the text the significance and p values.

In compliance with journal style and according to the reviewer’s comment, we have reduced the detailed text results to P-value. All figure legends were added with the text: Data is presented as mean \pm SEM.

15. The font used in the figures are not standardized, and some of them are too small, making difficult the understanding – such as Figure 1C, D; L-Q; Figure 3G, Figure 4L and N; Figure 6K on the right.

The font size was enlarged and standardized in all figures for clarity.

16. The format of the Western blot bands is different among figures. The suggestion is to follow the Figure 6A format, which is clear and organized.

All Western blot panels were adjusted according to the reviewer's comment.

References:

- 1 Muniz Moreno, M. D. M., Brault, V., Birling, M. C., Pavlovic, G. & Herault, Y. Modeling Down syndrome in animals from the early stage to the 4.0 models and next. *Progress in brain research* **251**, 91-143, doi:10.1016/bs.pbr.2019.08.001 (2020).
- 2 Rueda, N., Florez, J. & Martinez-Cue, C. Mouse models of Down syndrome as a tool to unravel the causes of mental disabilities. *Neural plasticity* **2012**, 584071, doi:10.1155/2012/584071 (2012).
- 3 Keren-Shaul, H. *et al.* A Unique Microglia Type Associated with Restricting Development of Alzheimer's Disease. *Cell* **169**, 1276-1290 e1217, doi:10.1016/j.cell.2017.05.018 (2017).
- 4 Krasemann, S. *et al.* The TREM2-APOE Pathway Drives the Transcriptional Phenotype of Dysfunctional Microglia in Neurodegenerative Diseases. *Immunity* **47**, 566-581 e569, doi:10.1016/j.immuni.2017.08.008 (2017).
- 5 Yin, Z. *et al.* Immune hyperreactivity of Abeta plaque-associated microglia in Alzheimer's disease. *Neurobiology of aging* **55**, 115-122, doi:10.1016/j.neurobiolaging.2017.03.021 (2017).
- 6 Jung, C. K., Keppler, K., Steinbach, S., Blazquez-Llorca, L. & Herms, J. Fibrillar amyloid plaque formation precedes microglial activation. *PloS one* **10**, e0119768, doi:10.1371/journal.pone.0119768 (2015).
- 7 Piatti, V. C. *et al.* The timing for neuronal maturation in the adult hippocampus is modulated by local network activity. *The Journal of neuroscience : the official journal of the Society for Neuroscience* **31**, 7715-7728, doi:10.1523/JNEUROSCI.1380-11.2011 (2011).
- 8 Ganesan, L. P. *et al.* The serine/threonine kinase Akt Promotes Fc gamma receptor-mediated phagocytosis in murine macrophages through the activation of p70S6 kinase. *The Journal of biological chemistry* **279**, 54416-54425, doi:10.1074/jbc.M408188200 (2004).
- 9 Garcia-Garcia, E. & Rosales, C. Signal transduction during Fc receptor-mediated phagocytosis. *Journal of leukocyte biology* **72**, 1092-1108 (2002).
- 10 Xu, P. *et al.* Microglial TREM-1 receptor mediates neuroinflammatory injury via interaction with SYK in experimental ischemic stroke. *Cell death & disease* **10**, 555, doi:10.1038/s41419-019-1777-9 (2019).

- 11 Ye, X. C. *et al.* Dectin-1/Syk signaling triggers neuroinflammation after ischemic stroke in mice. *Journal of neuroinflammation* **17**, 17, doi:10.1186/s12974-019-1693-z (2020).
- 12 Schweig, J. E. *et al.* Spleen tyrosine kinase (SYK) blocks autophagic Tau degradation in vitro and in vivo. *The Journal of biological chemistry* **294**, 13378-13395, doi:10.1074/jbc.RA119.008033 (2019).
- 13 Gitik, M., Kleinhaus, R., Hadas, S., Reichert, F. & Rotshenker, S. Phagocytic receptors activate and immune inhibitory receptor SIRPalpha inhibits phagocytosis through paxillin and cofilin. *Frontiers in cellular neuroscience* **8**, 104, doi:10.3389/fncel.2014.00104 (2014).
- 14 Mizuno, K. Signaling mechanisms and functional roles of cofilin phosphorylation and dephosphorylation. *Cellular signalling* **25**, 457-469, doi:10.1016/j.cellsig.2012.11.001 (2013).
- 15 Yoo, Y., Ho, H. J., Wang, C. & Guan, J. L. Tyrosine phosphorylation of cofilin at Y68 by v-Src leads to its degradation through ubiquitin-proteasome pathway. *Oncogene* **29**, 263-272, doi:10.1038/onc.2009.319 (2010).
- 16 Thom, G. *et al.* Isolation of blood-brain barrier-crossing antibodies from a phage display library by competitive elution and their ability to penetrate the central nervous system. *mAbs* **10**, 304-314, doi:10.1080/19420862.2017.1409320 (2018).
- 17 Villasenor, R. *et al.* Trafficking of Endogenous Immunoglobulins by Endothelial Cells at the Blood-Brain Barrier. *Scientific reports* **6**, 25658, doi:10.1038/srep25658 (2016).
- 18 Bruhns, P. & Jonsson, F. Mouse and human FcR effector functions. *Immunological reviews* **268**, 25-51, doi:10.1111/imr.12350 (2015).
- 19 Kiefer, F. *et al.* The Syk protein tyrosine kinase is essential for Fcgamma receptor signaling in macrophages and neutrophils. *Molecular and cellular biology* **18**, 4209-4220, doi:10.1128/mcb.18.7.4209 (1998).
- 20 Sierksma, A., Escott-Price, V. & De Strooper, B. Translating genetic risk of Alzheimer's disease into mechanistic insight and drug targets. *Science* **370**, 61-66, doi:10.1126/science.abb8575 (2020).
- 21 Yao, H. *et al.* Distinct Signaling Pathways Regulate TREM2 Phagocytic and NFkappaB Antagonistic Activities. *Frontiers in cellular neuroscience* **13**, 457, doi:10.3389/fncel.2019.00457 (2019).
- 22 Zhong, L. *et al.* TREM2/DAP12 Complex Regulates Inflammatory Responses in Microglia via the JNK Signaling Pathway. *Frontiers in aging neuroscience* **9**, 204, doi:10.3389/fnagi.2017.00204 (2017).
- 23 Xiang, X. *et al.* TREM2 deficiency reduces the efficacy of immunotherapeutic amyloid clearance. *EMBO molecular medicine* **8**, 992-1004, doi:10.15252/emmm.201606370 (2016).
- 24 Gratuze, M., Leyns, C. E. G. & Holtzman, D. M. New insights into the role of TREM2 in Alzheimer's disease. *Molecular neurodegeneration* **13**, 66, doi:10.1186/s13024-018-0298-9 (2018).
- 25 Ulland, T. K. & Colonna, M. TREM2 - a key player in microglial biology and Alzheimer disease. *Nature reviews. Neurology* **14**, 667-675, doi:10.1038/s41582-018-0072-1 (2018).
- 26 Sardesai, N. Y. & Weiner, D. B. Electroporation delivery of DNA vaccines: prospects for success. *Current opinion in immunology* **23**, 421-429, doi:10.1016/j.coi.2011.03.008 (2011).
- 27 Rosenberg, R. N., Fu, M. & Lambracht-Washington, D. Intradermal active full-length DNA Abeta42 immunization via electroporation leads to high anti-Abeta antibody levels in

wild-type mice. *Journal of neuroimmunology* **322**, 15-25,
doi:10.1016/j.jneuroim.2018.05.017 (2018).

REVIEWERS' COMMENTS:

Reviewer #1 (Remarks to the Author):

The authors made a thorough review of the manuscript, addressing all comments from reviewers. I really appreciate their openness in reformulating their approach to the results and discussion, which significantly improved the quality of the studies. This is a relevant and well-designed study, which is now suitable for publication.

Reviewer #2 (Remarks to the Author):

Manuscript Number: COMMSBIO-20-3006A

Title: "Maternal antibodies facilitate Amyloid- β clearance by activating Fc-receptor-Syk-mediated phagocytosis," by Illouz T. and colleagues.

The authors had the effort to respond to every question/suggestion in the manuscript's present version. Overall the work is improved, presenting an exciting and straight work at the technical and discussion levels.

Concerning the inclusion in the Introduction (from page 6) details on 5xFAD mice and the disadvantages as a DS model: although it is appreciated, the transfer of this text to the Discussion section is sufficient (text below).

The work will open interesting discussions regarding maternal immunization and neurodegenerative diseases.

Rebuttal letter

Maternal antibodies facilitate Amyloid- β clearance by activating Fc-receptor-Syk-mediated phagocytosis

2nd revision

Dear Dr. Montague-Cardoso,

Thank you for the opportunity to answer the reviewer's comments and provide a revised draft of our manuscript. We have addressed the reviewer's comments, as provided below.

Reviewer #1:

1. The authors made a thorough review of the manuscript, addressing all comments from reviewers. I really appreciate their openness in reformulating their approach to the results and discussion, which significantly improved the quality of the studies. This is a relevant and well-designed study, which is now suitable for publication.

Answer: We thank the reviewer for the positive critique of our work and for the insightful comments, contributing to the improvement of the manuscript.

Reviewer #2:

2. The authors had the effort to respond to every question/suggestion in the manuscript's present version. Overall the work is improved, presenting an exciting and straight work at the technical and discussion levels.

Concerning the inclusion in the Introduction (from page 6) details on 5xFAD mice and the disadvantages as a DS model: although it is appreciated, the transfer of this text to the Discussion section is sufficient (text below).

The work will open interesting discussions regarding maternal immunization and neurodegenerative diseases.

Answer: We thank the reviewer for the positive critique of our work. In accordance with this comment the following paragraph from the introduction section (page 6), was transferred to the discussion (page: 26, line: 18):

“Of note, the 5xFAD mouse model of EOAD used here does not provide a good model for DS in terms of construct validity, due to several reasons: (a) the model is based on *APP* and *PSEN1* mutations rather than on *APP* triplication, (b) the 5xFAD model lacks triplication of non-*APP* Hsa21 genes, and (c) the mutations in *PSEN1*, which is not characteristic of DS, may result in a confounding effect. Furthermore, the 5xFAD lacks tau-related mutations, and resultantly, no tau hyperphosphorylation and NFT formation occur in this model. Nonetheless, its face validity in recapitulating early A β plaque pathology, as occurs in DS, is satisfactory at the phenotype level compared with other DS mouse models.”